# Smart Homes and Families to Enable Sustainable Societies: A Data-Driven Approach for Multi-Perspective Parameter Discovery Using BERT Modelling

Eman Alqahtani [1] , Nourah Janbi [2] , Sanaa Sharaf [1] and Rashid Mehmood [3,*]

1   Department of Computer Science, Faculty of Computing and Information Technology,
    King Abdulaziz University, Jeddah 21589, Saudi Arabia
2   Department of Information Technology, College of Computing, and Information Technology at Khulais,
    University of Jeddah, Jeddah 21959, Saudi Arabia
3   High Performance Computing Center, King Abdulaziz University, Jeddah 21589, Saudi Arabia
*   Correspondence: rmehmood@kau.edu.sa

**Abstract:** Homes are the building block of cities and societies and therefore smart homes are critical to establishing smart living and are expected to play a key role in enabling smart, sustainable cities and societies. The current literature on smart homes has mainly focused on developing smart functions for homes such as security and ambiance management. Homes are composed of families and are inherently complex phenomena underlined by humans and their relationships with each other, subject to individual, intragroup, intergroup, and intercommunity goals. There is a clear need to understand, define, consolidate existing research, and actualize the overarching roles of smart homes, and the roles of smart homes that will serve the needs of future smart cities and societies. This paper introduces our data-driven parameter discovery methodology and uses it to provide, for the first time, an extensive, fairly comprehensive, analysis of the families and homes landscape seen through the eyes of academics and the public, using over a hundred thousand research papers and nearly a million tweets. We developed a methodology using deep learning, natural language processing (NLP), and big data analytics methods (BERT and other machine learning methods) and applied it to automatically discover parameters that capture a comprehensive knowledge and design space of smart families and homes comprising social, political, economic, environmental, and other dimensions. The 66 discovered parameters and the knowledge space comprising 100 s of dimensions are explained by reviewing and referencing over 300 articles from the academic literature and tweets. The knowledge and parameters discovered in this paper can be used to develop a holistic understanding of matters related to families and homes facilitating the development of better, community-specific policies, technologies, solutions, and industries for families and homes, leading to strengthening families and homes, and in turn, empowering sustainable societies across the globe.

**Keywords:** smart families; smart homes; sustainable societies; smart cities; natural language processing (NLP); Bidirectional Encoder Representations from Transformers (BERT)

## 1. Introduction

### 1.1. Home Sweet Home

Advancements in information and communication technologies (ICTs) and the consequent innovations have profoundly changed the lives of people in recent years, giving rise to smart environments, cities, and societies [1–4]. Technologies such as artificial intelligence (AI) and the Internet of Things (IoT) enhance quality of life for us by monitoring our environments and making decisions to achieve desirable outcomes. As homes are the building block of cities and societies, smart homes are critical to establishing smart living and are expected to play a key role in enabling smart cities and societies.

Homes, as is the case with many concepts, are interpreted differently by individuals. Després [5] discussed ten different meanings of homes, Gram-Hanssen and Darby [6] combined those ten meanings of homes into four categories, namely home as a place for "security and control", "activity", "relationships and continuity", and "identity and values". Mitty and Flores [7] defined homes as a physical location, a geographical location, or a place where meaningful relationships can be formed. Some see homes as places where control is important in determining the relationship between its members and their ability to make decisions [8]. Homes, therefore, may sound like a simple concept but the varying definitions of homes and the discussions around its various concepts show the complexity of the meanings of and concepts around homes.

A smart home is envisaged to consist of multiple network-connected devices, such as remote-controlled lighting, heating, kitchen, multimedia, and electronics appliances, usually integrated with sensors [9–12]. These sensors produce a large volume of data that is continuously analysed to allow people to monitor and engage with the environment and make intelligent decisions about their, safety, comfort, efficiency, etc. There is no universally agreed-upon definition of smart homes; however, there is a general understanding that smart homes are homes where various home-related functions are automated and enhanced through ICT technologies including IoT and AI. Homes are said to be 'smartened', or transformed into smart homes, by their automation through ICT technologies.

Considering the recent emphasis on sustainability and ethics in AI, we define smart homes as "homes that provide comfort, security, and other desirable features and meanings of homes, and make socially, environmentally, and economically sustainable and equitable decisions by using cutting-edge technologies, e.g., the Internet of Things (IoT), big data, artificial intelligence, and large-scale distributed cloud, fog, and edge computing". So, by this definition, smart homes provide various desirable features and meanings of homes, and in doing so, they make sustainable decisions that are regulated by the triple bottom line (TBL), i.e., the decisions are made to ensure social, environmental, and economic sustainability. We used the term 'equitable' in the definition to emphasize an important aspect of sustainability that concerns equity, explainability, ethics, and greening of AI, required to make equitable decisions [13,14]. In the past, heavy processing for smart homes was done in the cloud, causing long delays in making decisions by the system. With the rise of fog and edge computing, a large part of computing has moved close to homes (edge and fog) and this has reduced response time delays [15,16].

The current academic literature and commercial advancements on smart homes have mainly focused on developing and providing smart functions for homes to facilitate the residents in their various activities including ambiance management [17], energy management [18], smart appliance control [19], security management [20], and healthcare [9]. The fact that smart homes to date have focused on a limited set of home-related activities and functions could be seen in the findings of several recent literature reviews on smart homes; see e.g., [12,21–26]. For instance, our observation is confirmed by Gram-Hanssen and Darby [6] when they note, "The concept of home is largely absent from the thousands of papers in which building functions are analysed and modelled and the 'behaviours' of occupants are dissected and discussed".

## 1.2. This Work

This paper introduces our data-driven parameter discovery methodology and uses it to provide, for the first time, an extensive, fairly comprehensive, analysis of the families and homes landscape seen through the eyes of academics and the public using over a hundred thousand research papers and nearly a million tweets.

We combined deep learning, big data, and other technologies to create a complete machine learning pipeline for discovering parameters for families and homes from two different perspectives using two different types of data sources, namely academic literature on families and homes from the Scopus database and the public view from Twitter. The two types of data sources provide two distinct views of the families and homes domain,

one from academics and researchers and the other from the general public. These points of view are not mutually exclusive, and they influence each other to some extent, but they represent distinct perspectives with significant differences.

The purpose is to understand homes extensively, rather comprehensively and holistically, and use this understanding and knowledge to create an awareness of critical and other issues and drive future research on this topic using cutting-edge technologies. The parameters discovered and the knowledge gained in this research about families and homes could be used to direct smart homes research in areas that are important and have been neglected, fully or partially, in the past, leading to the enrichment of smart homes research and development of new technologies and industries. The ultimate aim is to develop a theory and practice that will lead to the development of smarter families and homes enabling sustainable future societies.

The Scopus database was used to create the academic-view dataset that we utilised to discover parameters for the academia-focused aspects of families and homes. We collected 104,018 research article abstracts with titles and keywords in English from a variety of academic disciplines including Social Science, Computer Science, Art and Humanities, and Multidisciplinary. The articles that we have collected were for the publishing period beginning from 2015 to the present. We discovered 44 parameters related to families and homes from the academic dataset and organized them into five macro-parameters, viz. Nurturing Families, Health & Lifestyles, Communities & Nations, Resources & Management, and Technologies.

The Twitter dataset that provides a public view of families and homes was collected for a period of six months, January to June 2022. A total of 930,110 tweets were retrieved. The data was limited to Saudi Arabia because we aimed to understand local issues related to families and homes and compare them with the international academic perspectives. We discovered 22 parameters and grouped them into three macro-parameters namely, Nurturing Families, Resources & Management, and Challenges.

We implemented the proposed data-driven approach for families & homes into a software tool. The tool consists of four software components: data collection, pre-processing, parameter modelling & discovery, and validation & visualisation. The tool can discover parameters related to families and homes using the datasets described above. The two datasets were collected and pre-processed to generate data in a form that the machine learning processing engine was able to process. We used a pre-trained BERT word embedding model, Bidirectional Encoder Representations from Transformers [27]. Subsequently, the Uniform Manifold Approximation and Projection (UMAP) [28] algorithm was used as a method for reducing dimensions, and Hierarchical Density-based Spatial Clustering of Applications with Noise (HDBSCAN) [29] was applied as a clustering algorithm. Moreover, the class-based TF-IDF score (term frequency-inverse document frequency) was used to cluster documents automatically in datasets [30]. We extracted two taxonomies from academic and public perspectives on families and homes. A range of quantitative methods was then used to analyze each dataset in order to discover families and homes parameters and macro-parameters, including similarity metrics [31], hierarchical clustering [32], term score [33], keyword score [34], and an inter-topic distance map [35]. The datasets, document clusters, and parameters were explored using various visualization methods, such as histograms [36], taxonomies, similarity matrices, temporal progression plots, and word clouds.

Figure 1 shows a multi-perspective (academic and public) view of families and homes comprising social, political, economic, environmental, and other issues discovered using Scopus and Twitter data. The figure exhibits the macro-parameters and parameters, laid out at the first branch level and the second branch level, respectively. The figure shows that the parameters discovered from the Scopus dataset provide a fairly comprehensive academic view of families & homes. The learned parameters capture and bring together, structurally, such wide-ranging issues including, among others, family roles and issues surrounding children, women and the elderly; specific physical and psychological dis-

eases; addictions; house pricing and affordability; architecture and heritage; work and employment; family businesses; farming; animal farms; tourism; energy management; water management; pandemics and disasters; technologies for remote, rehabilitation, and wearable-based healthcare and assistive robots; and issues specific to various political affiliations, races, communities, and nations from Canada, Europe, Asia, and America. The parameters detected by the Twitter data show a very local and public Saudi view of families with parameters such as an emphasis on nurturing family values; family cohesion; good companionship; gatherings with grandmothers; house financing and affordability; work and indolence; sleeping habits; and socioeconomic challenges for women. None of the earlier works have provided such an extensive view of families and homes captured systematically from academic and public perspectives. These could be used by, for instance, academia, to focus on problems that are more important for public and specific cultures, communities, and societies.

### 1.3. Novelty and Contributions

This paper's accomplishments are as follows. We developed a methodology using deep learning, natural language processing (NLP), and big data analytics methods and applied it to automatically discover parameters that capture a comprehensive knowledge and design space of smart families and homes comprising social, political, economic, environmental, and other dimensions. The discovered parameters and the knowledge space are explained by reviewing and referencing over 300 articles from the academic literature and tweets (see [37] for a more detailed account of the discovered parameters). We discovered 44 parameters for families and homes from an academic perspective that provide comprehensive structural knowledge and design space of families and homes that could be used to extend smart homes research and practice. We discovered 22 parameters for families and homes from the public perspective using Twitter data. This knowledge space could be used to develop culture and community-focused research and products. We compare the academic and public perspectives of families and homes, highlight the gaps, and propose a framework that discovers the knowledge and design space of families and homes, identifies parameters including opportunities, challenges, methods, solutions, and objectives and uses them to nurture better families and homes to enable smarter and sustainable societies. We built two datasets specifically for the work presented in this paper and created a complete big data analytics tool from the ground up for this purpose. The methodology, analysis, and the tool developed in this work are extensible and applicable to other subjects and topics, and the possibilities are endless.

The overarching goal of our research is to look into how ICT technologies can be used to solve pressing problems in smart cities and societies. Within the specific focus of this paper, we introduced in [38] the concept of Deep Journalism and discovered public, academic, and industry perspectives on transportation using The Guardian, Web of Science, and Traffic Technology International Magazine, respectively. We have also discovered parameters for education and learning during the COVID-19 pandemic [39] and healthcare services for cancer [40]. The work presented in this paper is novel for several reasons.

The current academic literature and commercial advancements on smart homes have mainly focused on developing and providing smart functions for homes to provide security management and facilitate the residents in their various activities, such as ambiance management. It is clear that the current efforts on smart home research and products are limited in their scope and do not consider the concept of home, particularly its social dimensions. There is a clear gap in understanding, defining, and in the actualization of the overarching roles of smart homes, and the roles of smart homes that will serve the needs of future smart cities and societies. This paper provides new approaches, methods, and findings and thereby fills in the above-mentioned gaps. None of the earlier works have provided such an extensive view of families and homes captured systematically from data bringing wide-ranging issues together. The novelty will be established further in Section 2 where we present the related works.

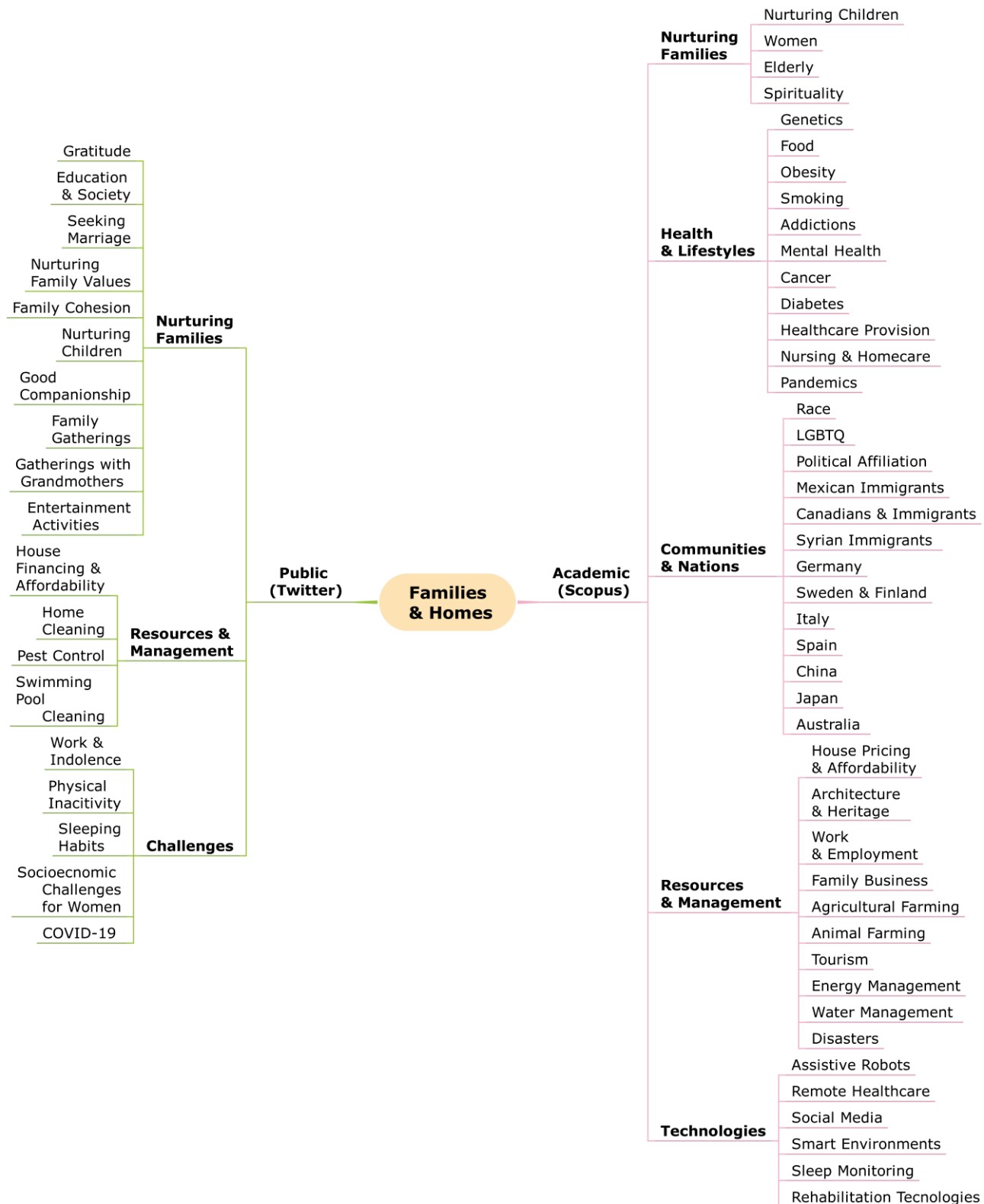

**Figure 1.** A multi-perspective taxonomy of Families & Homes.

The remainder of the paper is organized as follows. Section 2 discusses the works relevant to this paper and identifies the research gap. Section 3 describes the methodology of this work, which includes determining parameters for families and homes using Scopus and Twitter data, as well as the design of our tool. Section 4 explains and analyzes the

parameters discovered from Scopus data, offering an academic perspective on families and homes. Section 5 expands on the parameters discovered from Twitter data that provide a public view of families and homes. Section 6 contains a discussion, and Section 7 concludes with recommendations for future work. Figure 2 shows the map of Sections 3–5 to help the reader navigate through the article.

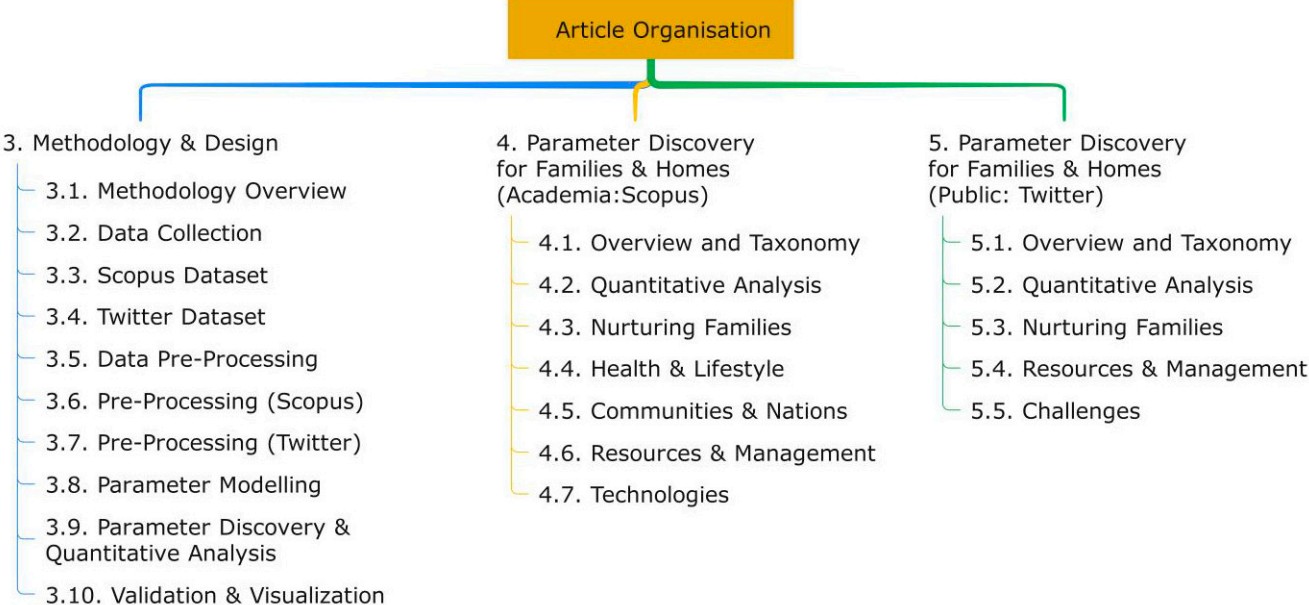

**Figure 2.** Article Organisation.

## 2. Related Works

In this section, we discuss works related to our proposed work. We conducted an extensive review of research on the use of AI and data analytics to understand families and homes and their related issues. We did not find any work directly related to the work proposed in this paper. The home is a very broad concept and is a transdisciplinary area of research, therefore, it is intricate to build a broad understanding of such a topic. However, to present the case of this work in the context of the overall body of works in this area, we present here related works from three areas: the meanings and concepts of homes, smart homes (technological aspects), and social media analytics (these works on social media are relevant because we use Twitter data to capture parameters for families and homes).

Firstly, we discuss the works on the meanings of home. Gram-Hanssen and Darby [6] have discussed the concept of smart homes from a broader view (meanings and concepts) and noticed the difference between the technical and conceptual literature on smart homes. Typically, technical research literature focuses on devices, tools, and technologies such as IoT and AI to provide a place for residents' activities, data security, and household control. However, the meaning of home goes beyond the walls and roof of a building and some activities to deeper aspects related to relations, values, identities, etc. which must be considered. They combined ten meanings of homes proposed by Després [5] into four categories, namely home as a place for "security and control", "activity", "relationships and continuity", and "identity and values". Gram-Hanssen and Darby [6] identified differences between the concepts of 'home' and 'smart home' in the literature but their main focus was on energy management research. Mitty and Flores [7] defined home as a physical location, a geographical location, or a place where meaningful relationships can be formed. Some see homes as places where control is important in determining the relationship between its members and their ability to make decisions [8].

Other studies have looked at the meaning of home for specific age groups (elderly, children, etc.) or specific cases (immigrants, some health conditions, etc.). For example, Hatcher et al. [41] focused on how older adults conceptualize home in light of age-related

lifestyle changes. Four major categories are identified to define the meaning of home for this age group: anchoring self, enabling freedom, being comfortable, and staying in touch. Alternatively, Lewin [42] explored the meaning of home for elderly immigrants and stressed the importance of considering age, gender, and cultural background.

Regarding smart home studies where researchers focus on the technological aspect of smart homes, multiple existing review papers have discussed smart home literature from different perspectives [12,21–26]. Regarding smart home definitions and characteristics, Marikyan et al. [22] suggested that the best way to facilitate the implementation and adoption of smart home technology would be by analysing the user's perspective and the current state. DeFrancoa and Kassaba [23] constructed a taxonomy for smart home research and noted that existing research avenues related to the concept of the smart home have not reached a consensus. Pira [24] discussed the social issues of smart homes and identified four main social barriers namely trust in controlling devices, service satisfaction, reliability of services, and privacy and security. Li et al. [12] concluded that the main research areas in smart homes are information and communication technologies (ICT) for home automation, home information management, AI for home automation, domestic energy management, and home-based health care.

For the smart Internet of Things (SHIoT) research, Choi et al. [21] identified the key dimensions as household, systems, network, and security. Topics under each of these dimensions were identified; for instance, the household dimension includes home automation, energy efficiency, domestic appliances, and intelligent buildings. Singh et al. [25] specifically explored the home health and internet of health things (IoHT) research and identified the assisted living of elderly patients using health monitoring devices as a key theme in this area. Two key themes are identified by Li et al. [26] for smart building research which are: (1) IoT, cloud computing, and wireless sensor network (WSN) for automation control and (2) balancing energy efficiency and human comfort using machine learning, and continuous monitoring.

The power of Twitter as an information source cannot be overstated. Many studies have used Twitter data in their research. For instance, researchers have conducted a thematic analysis of Twitter data in various study domains. For example, Alotaibi [2] introduced Sehaa, an Arabic-language big data analytics tool for healthcare in Saudi Arabia. A big data tool developed over Apache Spark, called Iktishaf, was proposed by Alomari [43,44] for detecting traffic-related events in Saudi Arabia based on Twitter data. Saurs et al. [45] used data mining techniques to identify the main security concerns in smart living environments. Many research have done on discovering COVID-19 issues using Twitter data analytics [46,47]. For instance, Su et al. [48] used Twitter to investigate the spatial-temporal factors and socioeconomic disparities that shaped U.S. residents' responses to COVID-19. An analysis of the sentiment and topics extracted from the COVID-19 tweets was conducted by Abdulaziz et al. [49]. Furthermore, Alswedani et al. [39] provided a comprehensive understanding of governance parameters related to the education sector using data-driven discovery tools developed for Twitter. Many other studies are available on the use of social media analytics on various topics [50–54]. However, as far as we know there is no study related to families & homes of a similar nature using Twitter data analytics in Arabic or in other languages.

## 3. Methodology & Design

This section explains the methodology and design of the proposed system architecture. Our software architecture is outlined in Figure 3, which consists of four software components that will be discussed in the following sections. Section 3.1 discusses the methodology overview, including the master algorithm. Sections 3.2–3.10 explain data collection, the data sources we used in this research (Scopus and Twitter), pre-processing, parameter modelling, parameter discovery & quantitate analysis, validation, and visualization, respectively.

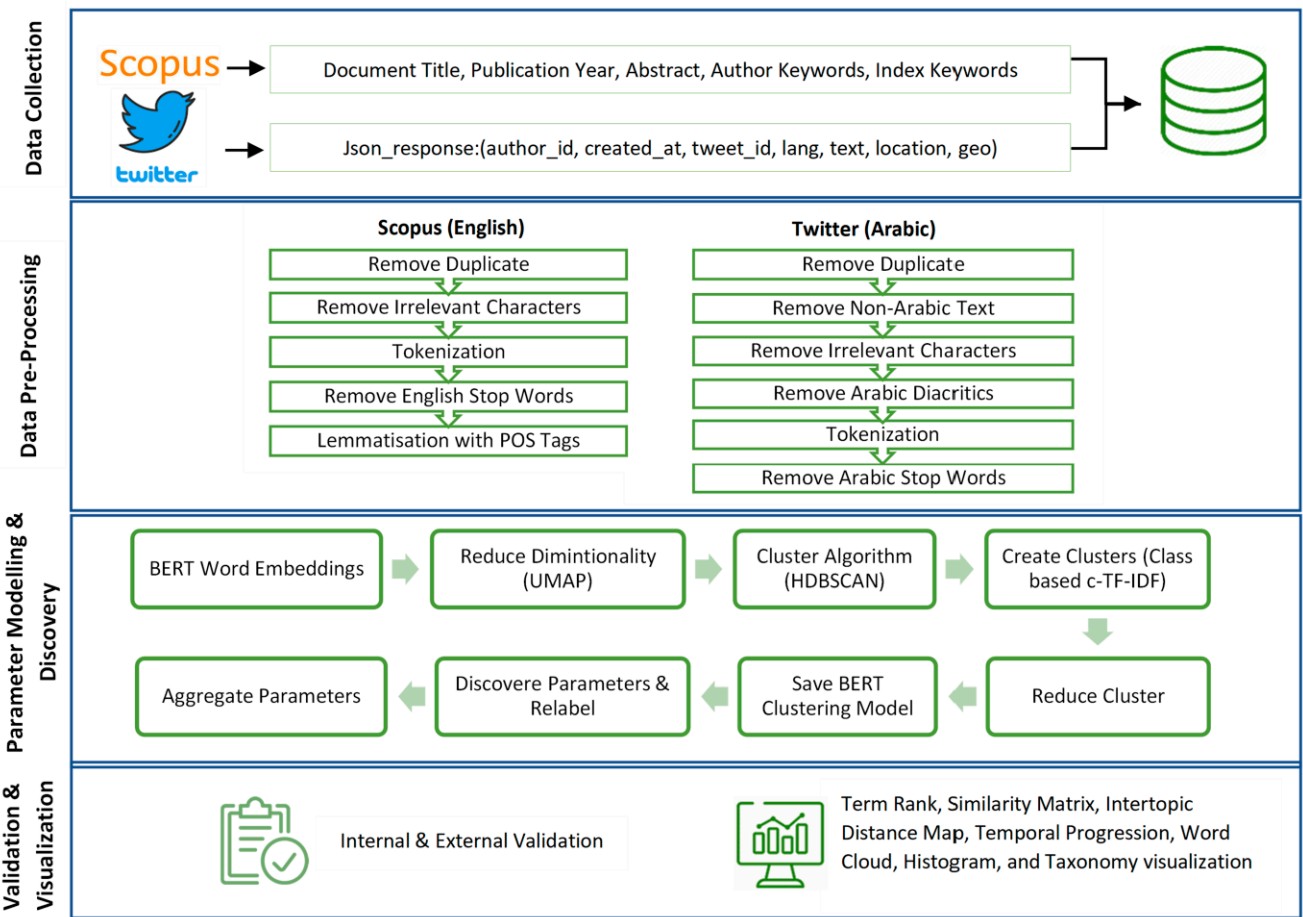

**Figure 3.** System Architecture.

### 3.1. Methodology Overview

Algorithm 1 provides a high-level master algorithm of our system. We built our dataset based on a specified search query then stored the results in a CSV file. Afterwards, the CSV file was loaded and pre-processed using Pandas. The contextual relationships between words were then captured with a pre-trained BERT word embedding model (Bidirectional Encoder Representations from Transformers) [27]. Subsequently, the UMAP (Uniform Manifold Approximation and Projection) [28] algorithm was used as a technique for reducing dimensions, and HDBSCAN (Hierarchical Density-based Spatial Clustering of Applications with Noise) was applied as a clustering algorithm [29]. Moreover, we merged similar clusters to reduce the number of clusters. Then we saved the clustering model. Based on the domain knowledge, similarity matrices, hierarchical clustering, and other quantitative analyses, the clusters were renamed as parameters, and ultimately the parameters were grouped into macro-parameters. Finally, we visualized the parameters and macro-parameters. In addition, we validated these parameters using two techniques: external and internal validation.

The BERTopic extracts a number of parameters from documents without requiring a prior definition of that number. Therefore, it is difficult to predict how many parameters will be extracted from our dataset before training our model. With the knowledge of the number of parameters generated, we specified a reasonable number of parameters using the reduction method, which merges the most similar parameters and then re-calculates the c-TF-IDF to update the representation of our parameters. Finally, the model was saved. Originally, the parameter was represented as an integer number. We used our domain knowledge and quantitative analysis of the keywords and documents in which we examined each parameter's keywords and looked at the context of the keywords in

each parameter. This process enabled us to re-label and give appropriate names to each parameter. The process was repeated iteratively to improve parameter names. The process also allowed us to remove irrelevant parameters and merge similar ones. We finally aggregated these parameters into macro-parameters representing higher-level families and home issues.

Our work was developed on Google Colab and Anaconda platforms. For data pre-processing and model training, we used python as the programming language along with libraries such as BERTopic [30], UMAP [28], HDBSCAN [29], Pandas [55], NumPy [56], NLTK [57], Scikit-Learn [58], and Gensim [59]. For data visualization, we used libraries such as Seaborn [60], Plotly [61], Matplotlib [62], and SciPy [63].

---

**Algorithm 1:** Master.

---

**Input:** Search Query
**Output:** The discovered Parameters & their Visualization
**1**　Collect data using the Search Query and Save to the Database
**2**　Read the saved data using Pandas Data Frame (data could be the English articles or the Arabic tweets)
**3**　Pre-process the Pandas DataFrame and return the Processed DataFrame
**4**　Train BERT model with the Processed DataFrame and generate the Word Embeddings
**5**　Reduce the dimensionality of the generated Word Embeddings using the UMAP algorithm
**6**　Cluster the Word Embeddings into groups of similar embeddings using the HDBSCAN algorithm
**7**　Extract Topics based on c-TF-IDF
**8**　Reduce the number of extracted Topics
**9**　Save the BERT model
**10**　Relabel the Topics as Parameters
**11**　Aggregate the Parameters as Macro-Parameters
**12**　Validate and Visualize the Parameters and Macro-Parameters

---

### 3.2. Data Collection

During our research, we used two data sources: Scopus (academic perspectives) and Twitter (public perspectives). Scopus is an indexing database of academic research; therefore, we consider that it provides an academic view of families and homes. Academic articles can describe public perspectives and situations; these perspectives are considered academic as they are perceived and expressed by academics (we understand that it depends on particular issues and situations and requires further elaborations). Twitter is a popular microblogging social media platform, and we used it to understand the public view of families and homes. Twitter can include posts from governments, industries and other stakeholders and hence can be used to understand other perspectives though tweets generated by various stakeholders that are generally used to engage with the public. The Scopus datasets were downloaded from the Scopus website in the CSV format. Twitter dataset was acquired using the Twitter API. Sections 3.3 and 3.4 discuss the data collection details for Scopus and Twitter, respectively.

### 3.3. Dataset (Scopus: English)

We obtained the most relevant documents from Scopus, one of the largest abstract databases covering a variety of scientific journals, conference proceedings, and books in various disciplines. The document types were limited to proceedings papers, articles, and reviews. We collected 104,018 research articles by using "Home", "House", and "Family" keywords with the OR logical relation from several different subject areas in Scopus: Social Science, Computer Science, Art, and Humanities, and Multidisciplinary. Furthermore, we have narrowed our search filtering option to the English language and, due to a large number of papers, limited the publishing years to between 2015 and the latest publication years (the latest publication dates could be 2023 or later years). Additionally, we utilized an advanced search and selected the "Topic Search" option, which yielded results from the document title, year, abstract, author keywords, and index keywords columns. Once the dataset had been collected under each subject area, we combined all the CSV files into a

single final CSV file. Any duplicate academic articles were removed before the final CSV file was passed to the next stage (pre-processing).

Figure 4 shows the histogram for the article abstracts. In this graph, the x-axis provides the word count of the articles in the dataset and the y-axis provides the number of articles in the dataset for a given number of words. Most article abstracts were between 300 and 400 words long. There were a few abstracts that were longer than 600 words.

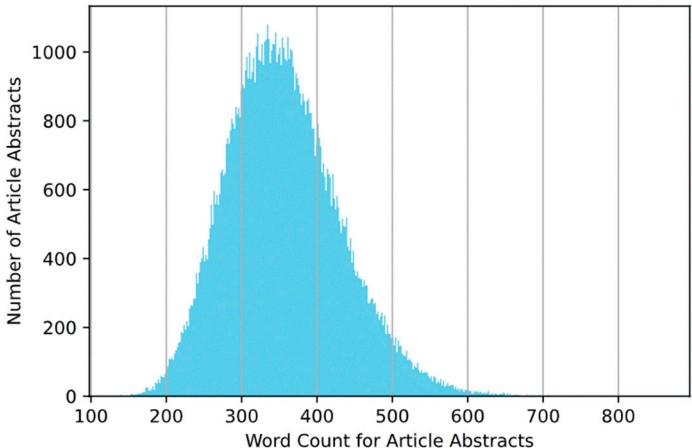

**Figure 4.** Histogram (Dataset: Scopus).

*3.4. Dataset (Twitter: Arabic)*

Twitter API V2 was used to collect the dataset from January 2022 to June 2022. The total number of retrieved tweets is approximately 930110. Initially, the tweets were acquired by using keywords and hashtags related to families & homes, such as "بيت" (Home), " منزل" (Manzel), " عائلة" (Family), "بيت#", "منزل#", and "عائلة#", and others. Using geolocation filtering, we obtained only tweets posted from Saudi Arabia because we aimed to find major families & homes issues related to Saudi society. Our collected Arabic tweets were retrieved from Twitter in JavaScript Object Notation (JSON) format. Each tweet includes several attributes such as "created_at" and "text", "geo" and "place" and others. Subsequently, we extracted and stored these attributes in a CSV file.

Figure 5 provides a histogram of the tweets in the dataset. The x-axis shows the number of tweets, and the y-axis shows the number of words in the tweets. The majority of tweets are approximately 20 to 50 words long. There are relatively few tweets that contain more than 75 words. The maximum number of words in a tweet are 120.

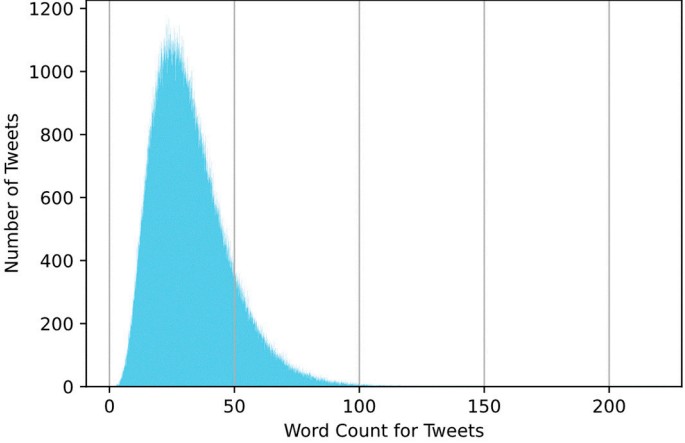

**Figure 5.** Histogram (Dataset: Twitter).

### 3.5. Data Pre-Processing

Data pre-processing is a crucial part of data analytics. In our research, we have used two types of data sources. A Scopus dataset contains English academic articles, while a Twitter dataset contains Arabic tweets. Due to the different languages used in these datasets, Scopus and Twitter cannot be pre-processed using the same algorithm. Sections 3.6 and 3.7 discuss the data pre-processing steps for Scopus and Twitter, respectively.

### 3.6. Pre-Processing (Scopus: English)

As part of the pre-processing step, we removed duplicate articles, irrelevant characters, tokenization, removed English stop words, then lemmatization with POS tags was performed. Initially, the CSV file was read by Pandas, a Python package, and saved in a data frame. Secondly, we encountered duplicate articles and eliminated them. Thirdly, we removed all irrelevant characters, including several Unicode characters. The step involved tokenizing the texts using the "Spacy" engine. Then, removal of the English stop words from the articles was done using the Natural Language Toolkit (NLTK) predefined list of stop words. As a final step, we lemmatized the data using the WordNetLemmatizer, which allows nouns, verbs, adjectives, and adverbs to be used as parts of speech. As a result of the pre-processing step, we obtained the cleaned articles and saved them in a CSV file thatwas used for parameter modeling and discovery.

### 3.7. Pre-Processing (Twitter: Arabic)

Our main pre-processing steps for the Twitter dataset were as follows: remove duplicate tweets, non-Arabic languages that use Arabic script, irrelevant characters, Arabic diacritics, tokenization, remove Arabic stop-words, then save cleaned tweets in a CSV file.

The CSV file was loaded in a data frame. Then, the duplicate tweets were removed using the Pandas package. Tweets written in non-Arabic languages that used Arabic scripts such as Urdu (ur), Persian (fa), and Central Kurdish (ckb) were also excluded. All English letters, numbers, mentions, and punctuation were removed. We also removed links and Arabic punctuation, including Arabic semicolons (؛) and The Arabic question mark (؟). In addition, Arabic diacritics have been removed where it categorized as a vowel, nunation, and shadda diacritics. The Arabic diacritics for short vowels include Fatha (◌َ), Damma (◌ُ), and Kasra (◌ِ), as well as Sukun diacritics (◌ْ), which indicate the absence of vowels. Arabic nunation diacritics represent the doubled versions of short vowels such as Fathatan (◌ً), Dammatan (◌ٌ), and Kasratan (◌ٍ). Shadda is the last form of diacritics (◌ّ) which can be combined with diacritics from the previous two types to create a new diacritic. NLTK stop-words as well as dialectical Arabic stop-words were excluded. Next, the Normalizer was used to normalize the words (tokens) that contain different forms of Alif (أ,إ,آ), Taa Marbutah (ة), and Yaa (ي) into the basic form. All these letters were replaced with the based form. For example, Alif was replaced by bare Alif (ا), Taa Marbutah was replaced by haa (ه), while Yaa was replaced by dotless Yaa (ى). The text was tokenized as the next step using the Spacy package.

### 3.8. Parameter Modelling

As the first step to parameter modeling, we developed a word-embedding model using a technique developed by Google called BERT (Bidirectional Encoder Representations from Transformers) [27]. Textual data was analyzed using BERT to extract features, including word embeddings and sentence embeddings. Our paper used the pre-trained "distilbert-base-nli-mean-tokens" model, which is a model of sentence-transformers. It can be used for tasks like clustering or semantic search by mapping sentences and paragraphs into a dense vector space with 768 dimensions. In terms of performance, it offers a nice balance between speed and efficiency. In addition, there are several multi-lingual models available in the package. For keeping maximum information in a lower dimension, we implemented

a dimensional reduction algorithm called UMAP. It has several parameters, but the most important are n_neighbors and n_components. The n_neighbors parameter controls how UMAP balances local and global data structures. In addition, it defines the neighborhood size, low-value focus locally, and high-value focus globally. The n_components parameter controls the dimensionality to find the embed data. There are no obvious ways to choose the best values, depending on the situation. According to Angelov's paper [64], the best is n_neighbors = 15 and n_components = 5. We grouped similar articles to define a cluster or parameter using the HDBSCAN algorithm. The most important HDBSCAN parameters are min_cluster_size and min_sample. The min_cluster_size parameter controls the smallest cluster. The min_sampling parameter controls the cluster size, and when min_sample is smaller than min_cluster_size, it will merge that article to the same cluster. More articles are discarded when the min_sampling is high. Since UMAP retains local structures at lower dimensions, HDBSCAN complements UMAP and does not force outlier articles to a cluster.

Additionally, the c-TF-IDF score (Class Based Term Frequency-Inverse Document Frequency) is used to calculate words' importance for each document. The TF-IDF score (Term Frequency-Inverse Document Frequency) determines the frequency and prominence of a word in each document and evaluates the word relevance between documents. While the c-TF-IDF considers all the documents of a cluster as a single whole document and then executes the TF-IDF on it, it will achieve a significance score for each word inside a cluster called the c-TF–IDF score. Words that are most significant within a cluster represent that cluster most effectively. In this way, we were able to obtain keyword-based descriptions for every cluster, which helped to specify the parameter name for it. A c-TF-IDF score was calculated using Equation (1) [34], where f = the word frequency is derived for each class or cluster c and divided by the number of words w. The total number of un-joined documents (d) was then divided by the total frequency of words (f) throughout all classes (cc).

$$\mathrm{c} - \mathrm{TF} - IDF_C = \frac{\mathrm{f_c}}{w_c} \times \log \frac{\mathrm{d}}{\sum_{\mathrm{p}}^{\mathrm{cc}} \mathrm{f}_p} \tag{1}$$

The first part of the equation calculates the Term Frequency (TF) score per class. It can be seen as regularizing frequently used words in a class. The second part determines how common a term is based on its Inverse Document Frequency (IDF) score. The relevance of rare terms is greater than that of common terms. With both TF and IDF scores for each term (word), we can multiply the two to calculate the c-TF-IDF score.

Predicting how many parameters will be extracted from our documents is difficult before training our model. Therefore, BERTopic was trained on our documents, which resulted in several parameters. After knowing how many parameters were created, we decided and specified a reasonable number of parameters using parameter reduction. After that, all parameters were assigned to the articles, and the model was saved.

BERTopic

BERTopic uses class-based TF-IDFs and BERT embeddings to create dense clusters. BerTopic supports a variety of modeling methods, including hierarchical, guided, semi-supervised, dynamic, and online methods. Many embedding models can be used with BERTopic, including Sentence-Transformers, Transformers, Flair, Spacy, Gensim, and USE. The sentence-Transformers model has shown great results in embedding documents with semantic similarity, so we use it here. Different visualization methods are supported, such as word scores, inter-topic distance map, hierarchical clusters, and similarity matrix, which help to understand the model and make changes as needed. BERTopic can extract the number of topics described in documents and does not require a prior definition of the number of topics. This is one of the advantages of BERTopic over the most popular topic modeling techniques such as Latent Dirichlet allocation (LDA) and Non-Negative Matrix Factorization (NMF). Further information can be found in these resources [27,30,64,65]:

Originally, the parameter was represented as an integer number, and we used our domain knowledge and quantitative analysis methods to re-label and aggregate it into macro-parameters. We discuss this in the following section.

### 3.9. Parameter Discovery & Quantitative Analysis

To get a good understanding of the extracted topics, we can iteratively go through perhaps a hundred topics after training our BERTopic model. Nonetheless, this takes a considerable time and does not offer a global view. A better approach would be to visualize the topics generated using quantitative analysis methods. Therefore, we determined and understood the parameters and macro-parameters for this study from quantitative analysis methods and our domain knowledge, such as term scores, inter-topic distances, keyword scores, and hierarchical clustering.

#### 3.9.1. Term Score

It is not easy to express the context of a parameter based on a list of keywords (terms). Our first step in finding a parameter was determining how many keywords were needed and the starting and ending positions of significant keywords. The keywords c-TF-IDF score for each parameter were visualized by sorting them in decreasing order. This term score visualization greatly influences parameter identification [30].

#### 3.9.2. Inter-Topic Distance Map

The inter-topic distance map represents the parameters in a two-dimensional way, represented by parameter circles whose size corresponds to the number of words used in the dictionary to describe that parameter. A MinMaxScaler algorithm is used to form the circles. The parameters closer together share more words [30].

#### 3.9.3. Keyword Score

BERT parameter models generate a list of keywords that describe a parameter, each having an importance score or c-TF-IDF (see Section 3.8) for contextualizing the parameter [30].

#### 3.9.4. Hierarchical Clustering

Clusters are systematically paired to create hierarchical clusters using the cosine similarity matrices between the parameter embeddings [30]. Starting with the correlation matrix in each phase, all possible pairs of clusters are tried, and the pair with the largest average correlation within the experimental cluster is chosen as the new unique cluster.

#### 3.9.5. Similarity Matrix

Plotly, a Python library is used to visualize the similarity matrix between parameters based on the cosine similarity matrix [30]. We calculated the cosine similarity score between the parameters embedding to show the relationship between the parameters. According to Plotly "BnGu" (green to blue), the dark blue color represents the highest relationship between parameters, and the light green color represents the lowest similarity relationship.

### 3.10. Validation & Visualization

Results can be validated internally and externally. Internal validation of a parameter involves investigating and discussing the documents related to the parameter. In our research, documents could be academic articles or tweets. We discussed how we perceived the correlation between the documents and the parameters in most of the documents in our dataset. External validation is done by comparing the two datasets' parameters, keywords, and metric metrics. For the visualization various visualization methods are used for the internal and external validation. Many visualization methods are used to describe the datasets, the clusters of documents, and the parameters that have been discovered. These are dataset histograms [36], taxonomies, similarity matrices [66], term rank,

similarity matrix, inter-topic distance map, temporal progression plots, and word clouds. These visualizations are created using several Python libraries, including Seaborn, Plotly, and Matplotlib.

## 4. Parameter Discovery for Families & Homes (Academia: Scopus)

This section discusses the parameters detected by our BERT model from the Scopus dataset. Section 4.1 provides an overview of the parameters and macro-parameters. Section 4.2 provides quantitative analysis of the clustering characteristics and discovered parameters. In the subsequent sections, Sections 4.3–4.7 we discuss each individual macro-parameter in detail. The temporal analysis of the parameters and macro-parameters is presented in Section 4.7.7.

### 4.1. Overview and Taxonomy

The modeling process detected 50 clusters from the Scopus dataset using the BERT modeling algorithm. We excluded six parameters from the initial clustering results as they were irrelevant to the topic of this work. Four clusters captured parameters related to the families of Animals (Clusters 12 and 40) and Plants (Clusters 11 and 32) due to the keyword "families" in the article search. Software Development (Cluster 1) and Miscellaneous (Cluster 6) were the other two irrelevant characters. Note that our approach in this paper is to cluster the data and remove irrelevant clusters. Another approach could be to filter irrelevant data before clustering. Nevertheless, another approach could be to find irrelevant clusters, use them to filter the dataset, and perform clustering again on the filtered data. These alternative approaches will be investigated in the future.

The remaining 44 clusters, called parameters, were grouped into five macro-parameters based on domain knowledge, similarity matrix, hierarchical clustering, and other quantitative methods. The methodology and process used to discover parameters and group them into macro-parameters have already been described in Section 3.

Table 1 lists some information about these parameters. The parameters, their numbers in the clustering model, and macro-parameters are listed in Columns 1 to 3, respectively. Some of the parameters have been merged. The percentage of the articles for each parameter is listed in Columns 4 and 5. Our BERT model labeled 47.01% of the articles with the outlier cluster; therefore, the total percentage of articles listed in the table is 52.99%. The sixth column shows the top 20 keywords associated with each parameter sorted according to their importance score.

Figure 6 provides a taxonomy of Families & Homes domain extracted from academia. The taxonomy is created using the parameters and macro-parameters discovered from the Scopus dataset. The macro-parameters are shown on the first level of branches, and the discovered parameters are shown on the second level of branches.

**Table 1.** Macro-Parameters and Parameter for Families & Homes (Data Source: Scopus).

| Macro | Parameter | ID | % | Keywords |
|---|---|---|---|---|
| Nurturing Families | Nurturing Children | 0 | 8.33 | child, student, parent, school, family, study, education, teacher, language, learn, home, social, research, parental, experience, educational, academic, year, relationship, literacy |
| | Women | 4 | 4.24 | woman, gender, mother, health, female, pregnancy, family, work, study, social, maternal, article, birth, child, adult, violence, life, contraceptive, home, research |
| | Elderly | 7 | 1.23 | old, elderly, old adult, home, live, adult, old people, health, life, social, elderly people, fall, daily, study, support, sensor, caregiver, aged, home care, base |
| | Spirituality | 23 | 0.40 | religious, religion, church, spirituality, christian, chaplain, spiritual care, muslim, prayer, faith, theology, community, study, protestant, theological, catholic, law, secular, patient, marriage |

**Table 1.** *Cont.*

| Macro | Parameter | ID | % | Keywords |
|---|---|---|---|---|
| Health & Lifestyles | Genetics | 16 | 0.52 | mutation, genetic, gene, protein, disease, pedigree, phenotype, patient, study, exome sequence, female, clinical, nucleotide, autosomal, male, dna, genotype, protein human, polymorphism, novel |
| | Food | 15 | 0.58 | food, eat, meal, nutrition, food waste, household, restaurant, study, school, consumer, diet, dietary, health, vegetable, grocery, food insecurity, nutritional, food consumption, cooking, food security |
| | Obesity | 49 | 0.19 | obesity, overweight, overweight obesity, body mass, weight, childhood obesity, mass, prevalence, health, mass index, index, obese, adult, age, female, physical, family, high, male, adolescent |
| | Smoking | 41 | 0.23 | smoking, tobacco, cigarette, smoker, smoke free, tobacco product, smoking cessation, adolescent, quit, study, adult, tobacco use, nicotine, home, health, secondhand, shs exposure, passive smoking, female, free |
| | Addictions | 44 | 0.22 | alcohol, drink, gambling, drinking, drug, overdose, opioid, adolescent, naloxone, alcohol use, consumption, cocaine, male, study, alcohol consumption, female, family, alcoholic, health, disorder |
| | Mental Health | 30 | 0.33 | dementia, caregiver, people dementia, live, alzheimer, dementia care, live dementia, home, dementia family, family, patient, person dementia, disease, health, family caregiver, aged, nursing, staff, alzheimer disease cognitive, |
| | Cancer | 3 | 4.49 | protein, cell, cancer, gene, mouse, tumour, animal, receptor, kinase, molecular, drug, enzyme, family, cell line, breast, genetic, domain, gene expression, acid, DNA |
| | Diabetes | 38 | 0.26 | diabetes, type diabetes, glucose, insulin, patient, blood, diabetic, study, health, diabete, insulin dependent, disease, dependent diabetes, adult, risk factor, family history, female, non insulin, male, history |
| | Healthcare Provision | 46 | 0.19 | student, medical, nursing, medical student, education, nursing student, clinical, undergraduate, patient, medical education, health, nurse, medical school, medicine, education medical, practice, nursing education, study education nursing, interprofessional, |
| | Nursing & Homecare | 9 | 1.02 | patient, hospital, study, nurse, death, home, health care, nursing, aged, clinical, healthcare, medical, female, disease, medication, tuberculosis, physician, nursing home, article, family |
| | Pandemics | 22 | 0.40 | vaccine, covid, infection, virus, disease, vaccination, coronavirus, pandemic, influenza, viral, epidemic, spread, respiratory, study, outbreak, infant, model, patient, pneumonia, social |
| Communities & Nations | Race | 14 | 0.62 | black, racial, race, african, racism, family, health, hispanic, social, black woman, american, woman, slavery, segregation, child, black, white, african american, racial ethnic, black family, research |
| | LGBTQ | 28 | 0.37 | gay, transgender, bisexual, sex, male, heterosexual, lesbian, homosexuality, parent, adult, health, gay bisexual, gay man, social, study, queer, sexual gender, female, youth, father |
| | Political Affiliations | 17 | 0.51 | party, election, political, vote, electoral, voter, presidential, candidate, house, partisan, president, parliamentary, voting, parliament, democracy, congressional, democratic, trump, legislator, political party |
| | Mexican Immigrants | 48 | 0.19 | mexican, mexican origin, immigrant, parent, adolescent, migrant, migration, study, mother, familism, health, mexican immigrant, female, experience, social, transnational, human, adult, policy, mexican american |
| | Canadians & Immigrants | 20 | 0.44 | canadian, family, study, child, social, indigenous, policy, community, health care, research, home care, practice, mental, adult, article, life, patient, physician, mental health, immigrant |
| | Syrian Immigrants | 45 | 0.22 | refugee, syrian, immigration, deportation, asylum, syrian refugee, migration, migrant, family, country, resettlement, displacement, camp, child, detention, mental health, home, asylum seeker, interview, article |
| | Germany | 25 | 0.39 | german, family, war, home, study, article, social, life, history, germany, child, work, research, author, migration, country, century, health, refugee, memory |

**Table 1.** *Cont.*

| Macro | Parameter | ID | % | Keywords |
|---|---|---|---|---|
| Communities & Nations | Sweden & Finland | 29 | 0.37 | swedish, sweden, finland, finnish, home, family, child, study, social, second home, article, health, parent, life, policy, language, school, home care, education, parental |
| | Italy | 31 | 0.32 | italian, family, social, study, work, child, migrant, language, article, history, parent, home, migration, house, immigrant, lockdown, gender, economic, life, author |
| | Spain | 39 | 0.25 | spanish, english, bilingual, spanish english, family, vocabulary, spanish speak, study, heritage, speaker, literacy, home, parent, home language, proficiency, linguistic, read, dual language, social, learner |
| | China | 19 | 0.44 | chinese, family, child, study, social, housing, chinese family, home, market, parent, policy, culture, immigrant, migrant, filial, firm, government, economic, relationship, research |
| | Japan | 42 | 0.22 | japanese, care, language, family, study, home, child, work, caregiver, culture, article, life, health, survey, medical, student, old, house, adult, social |
| | Australia | 8 | 1.16 | australian, australia, homelessness, housing, homeless, home, health, social, study, parent, young, research, mental, community, aboriginal, work, policy, article, school, interview |
| Resources & Management | House Pricing & Affordability | 18 | 0.48 | price, house price, housing, house, market, real estate, estate, housing price, model, housing market, hedonic, prediction, urban, price prediction, mortgage, datum, residential, land, income, buyer |
| | Architecture & Heritage | 21 | 0.44 | century, house, archaeological, date, late, neolithic, excavation, early, bronze, pottery, bronze age, burial, evidence, medieval, history, archaeology, ancient, chronology, radiocarbon, iron |
| | Work & Employment | 26 | 0.39 | work, employee, work family, job, family, family conflict, worker, work home, workplace, job satisfaction, study, life, stress, relationship, resource, organizational, work life, supervisor, employment, home |
| | Family Business | 34 | 0.28 | business, family firm, family, family business, corporate, company, governance, non family, social responsibility, corporate governance, corporate social, family ownership, ceo, study, social, financial, sustainability, firm performance, relationship, innovation |
| | Agricultural Farming | 24 | 0.40 | farmer, farm, agricultural, land, family farm, agriculture, farming, food, rural, crop, household, smallholder, economic, market, labor, sustainable, study, adoption, local, climate |
| | Animal Farming | 13 | 0.63 | dog, home range, range, habitat, pet, chicken, cat, poultry, bird, specie, pig, home, wild, female, male, meat, owner, article, house, animal |
| | Tourism | 33 | 0.28 | tourism, tourist, hotel, travel, guest, holiday, tourist destination, family, sustainable, airbnb, research, cruise, accommodation, social, second home, local, house, tourism industry, heritage, business |
| | Energy Management | 5 | 3.09 | energy, grid, electric, management, electricity, building, solar, energy management, air, appliance, renewable, residential, house, home energy, thermal, battery, indoor, heating, renewable energy, energy consumption |
| | Water Management | 37 | 0.26 | water, water supply, drinking water, sanitation, water quality, household, drinking, supply, water consumption, urban, water treatment, drink water, rainwater, water source, water use, water management, sensor, wash, household water, toilet |
| | Disasters | 27 | 0.38 | flood, disaster, earthquake, hurricane, evacuation, coastal, landslide, community, tsunami, flooding, flood risk, impact, event, natural disaster, recovery, household, study, storm, housing, resilience |
| Technologies | Assistive Robots | 10 | 0.88 | robot, robotic, home, human robot, robot interaction, design, service robot, environment, mobile robot, user, mobile, social robot, assistive, base, social, machine, propose, intelligent, paper, technology |
| | Remote Healthcare | 47 | 0.19 | health, patient, healthcare, medical, internet, datum, sensor, internet thing, monitor, remote, health monitoring, security, wearable, network, health care, home, cloud, technology, mobile, privacy |

**Table 1.** *Cont.*

| Macro | Parameter | ID | % | Keywords |
|---|---|---|---|---|
| Technologies | Social Media | 43 | 0.22 | social medium, twitter, facebook, online, user, networking, social networking, tweet, social network, network, news, networking online, youtube, friend, online social, broadband, internet, study, digital, video |
| | Smart Environments | 2 | 6.27 | home, smart home, internet, network, security, internet thing, sensor, automation, wireless, base, technology, intelligent, propose, malware, datum, detection, home automation, attack, paper, mobile |
| | Sleep Monitoring | 35 | 0.28 | sleep, apnea, sleep quality, sleep apnea, sleep stage, night, polysomnography, sleep disorder, disorder, sleep duration, stage, study, home, sleep, sleep research, wake, adolescent, patient, sleep monitor, male |
| | Rehabilitation Technologies | 36 | 0.27 | rehabilitation, exercise, patient, parkinson, parkinson disease, game, stroke, training, disease, home, base, therapy, motor, virtual, home base, pd, week, feedback, wearable, upper limb |

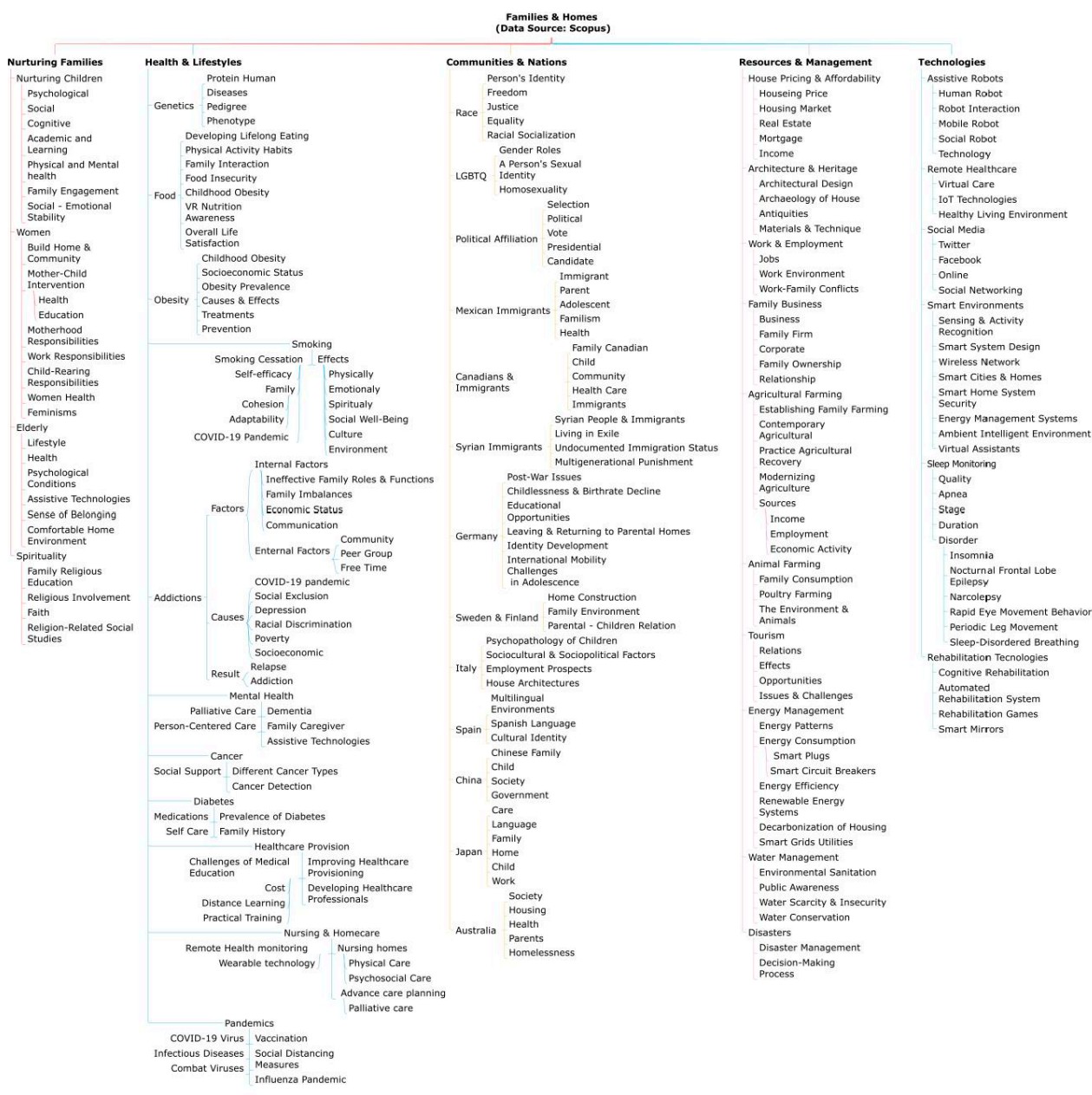

**Figure 6.** A Taxonomy of Families and Homes Extracted from the Scopus Dataset.

### 4.2. Quantitative Analysis

This section discusses term and word scoring, inter-topic distance mapping, hierarchical clustering, and similarity matrices. A group of keywords represents almost all parameters; not all of them are equally descriptive. Figure 7 indicates the number of keywords needed to describe a parameter and at what point the benefit of adding more keywords diminishes (see Section 3.9). Note in the plots that only the top seven to ten terms in each parameter accurately describe that parameter. Since all the other probabilities are close to each other, it makes no sense to rank them. Therefore, to label the parameter, we focused on that parameter's top seven to ten keywords.

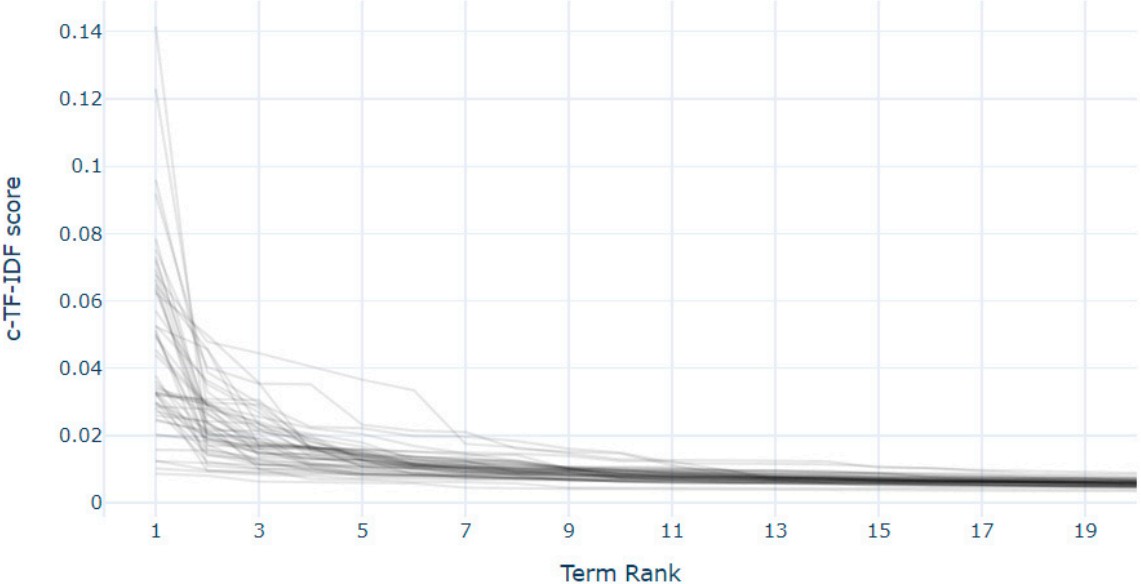

**Figure 7.** Term Rank (Data Source: Scopus).

The top ten keywords for each parameter are shown in Figure 8 Families and Homes Parameters with Keywords c-TF–IDF Score Figures 8 and 9. The importance score c-TF-IDF is used to order the keywords (see Section 3.9). There are 44 bar charts (bars are in different colors to help differentiate between them and the colors do not have any specific meaning or representation), where the horizontal line indicates the importance score, and the vertical line indicates the parameter keywords.

Figure 10 shows the inter-topic distance map, where twelve groups of parameters are identified (see Section 3.9). The BERT model does not specify exact name to each cluster rather it represent an integer number along with 'Topic' term for example: Topic 0 refers to cluster 0 and so on. Therefore, we re-labeled them using our domain knowledge and quantitative analysis information. Then, manual grouping of the parameters into five macro-parameters is also performed. D1 and D2 show the two dimensions.

Figure 11 describes the automated hierarchical clustering of the 50 clusters. It systematically pairs them based on the cosine similarity matrix (see Section 3.9).

Figure 12 visualizes the similarity matrix among the parameters (see Section 3.9). The dark blue color represents the highest similarity between parameters, whereas light green represents the lowest similarity. For example, Cluster 10, labeled as Assistive Robots, and Cluster 47, labeled as Remote Healthcare, have high similarity scores as they are both important technologies that focus on caring for the patient; robotics can be used to provide high-quality patient care in the healthcare field.

### 4.3. Nurturing Families

The Nurturing Families macro-parameter concerns about strengthening families and homes and the associated challenges. It comprises four parameters, namely, Nurturing Children, Woman, Elderly, and Spirituality.

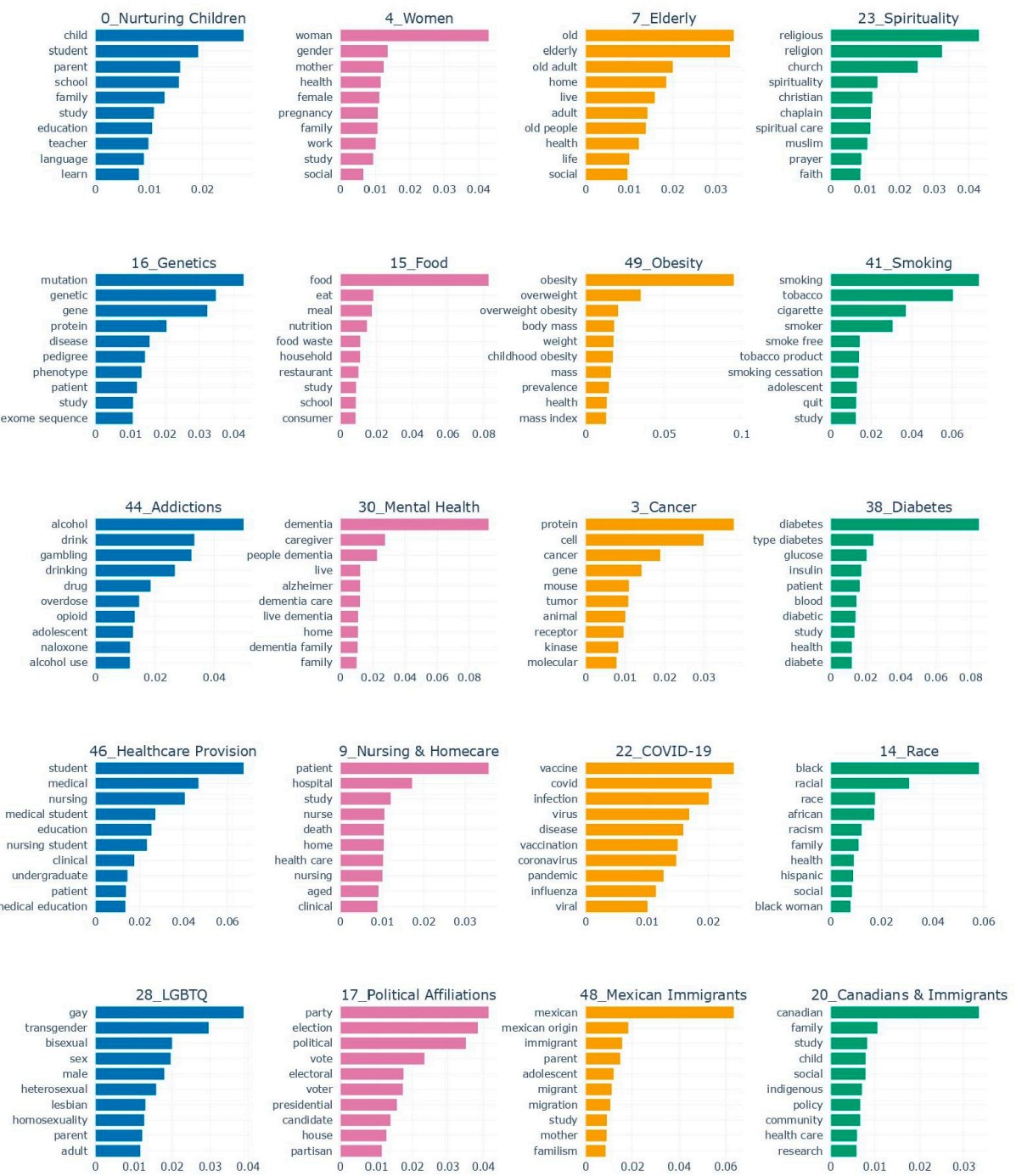

**Figure 8.** Families and Homes Parameters with Keywords c-TF–IDF Scores (Data Source: Scopus) (Part A).

### 4.3.1. Nurturing Children

The Nurturing Children parameter focuses on the development of children's academic abilities and social skills, among other issues, along with the involvement of parents and teachers. The keywords that our model detected include child, student, parent, school, family, study, education, teacher, language, learn, home, social, research, parental, experience, educational, academic, relationship, and literacy. Taking a look at the academic articles that relate to this parameter, we were able to find a number of topics that describe various dimensions of this parameter. The topics and dimensions include psychological [67–69],

social [70], cognitive [71], physical and mental health development [72–74], family engagement [75–77], behaviour change [78,79], and education [70,80–82].

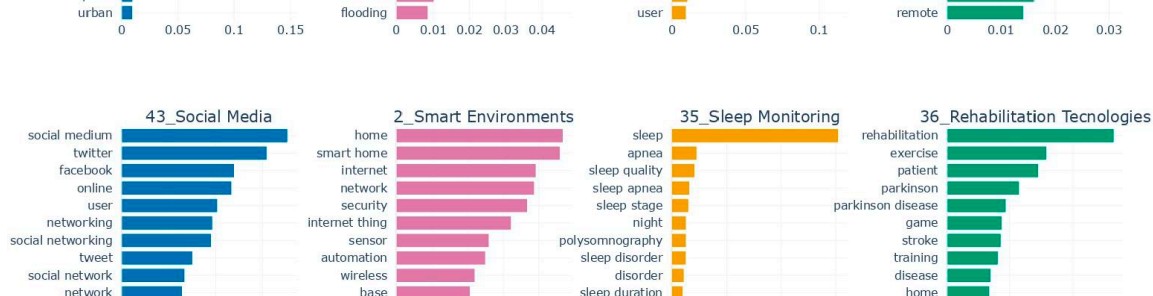

**Figure 9.** Families and Homes Parameters with Keywords c-TF–IDF Scores (Data Source: Scopus) (Part B).

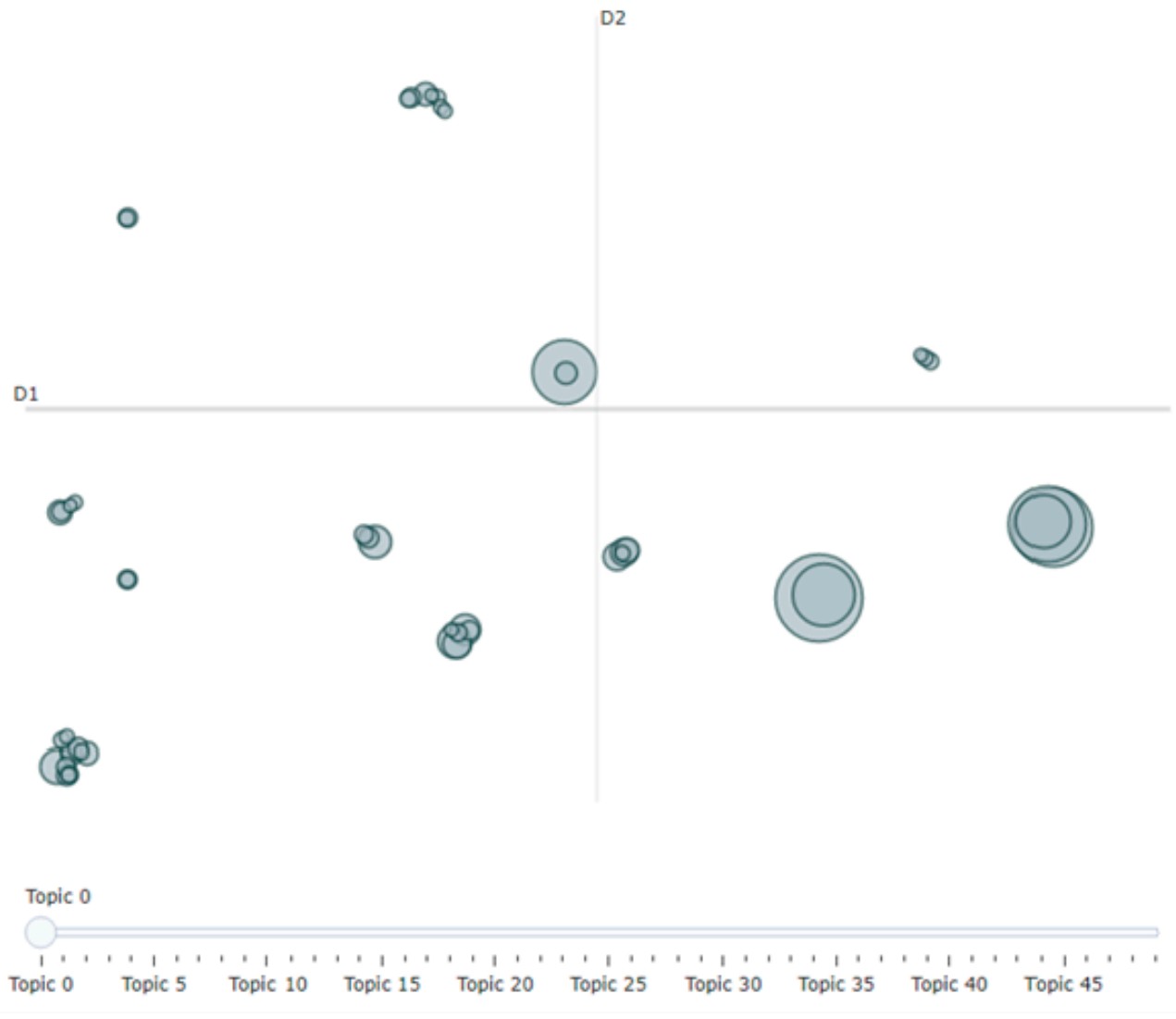

**Figure 10.** Inter-topic distance map (Data Source: Scopus).

4.3.2. Woman

The Woman parameter covers different roles women play in the families & homes, challenges, issues, advice, and technological enablers while developing, maintaining, and taking care of their families. In addition, documents under this parameter discussed some social movements related to women such as feminism. The keywords that were detected by our model include woman, gender, mother, health, female, pregnancy, family, work, study, social, maternal, article, birth, child, adult, violence, life, contraceptive, home, and research.

Tara Beagan in her book "Honour Beat" described the mother as the home and community and stated that they are synonymous [83]. Shai [84] stated that in order to strengthen women's roles in the household, young women in communities need to be protected from violence and discussed family-centered intervention methods to prevent violence against girls and women. Smith [85] focused on the eighteenth century and affirmed the important role that women played in Britain's elite imperial families in home building in terms of organizing and completing the work.

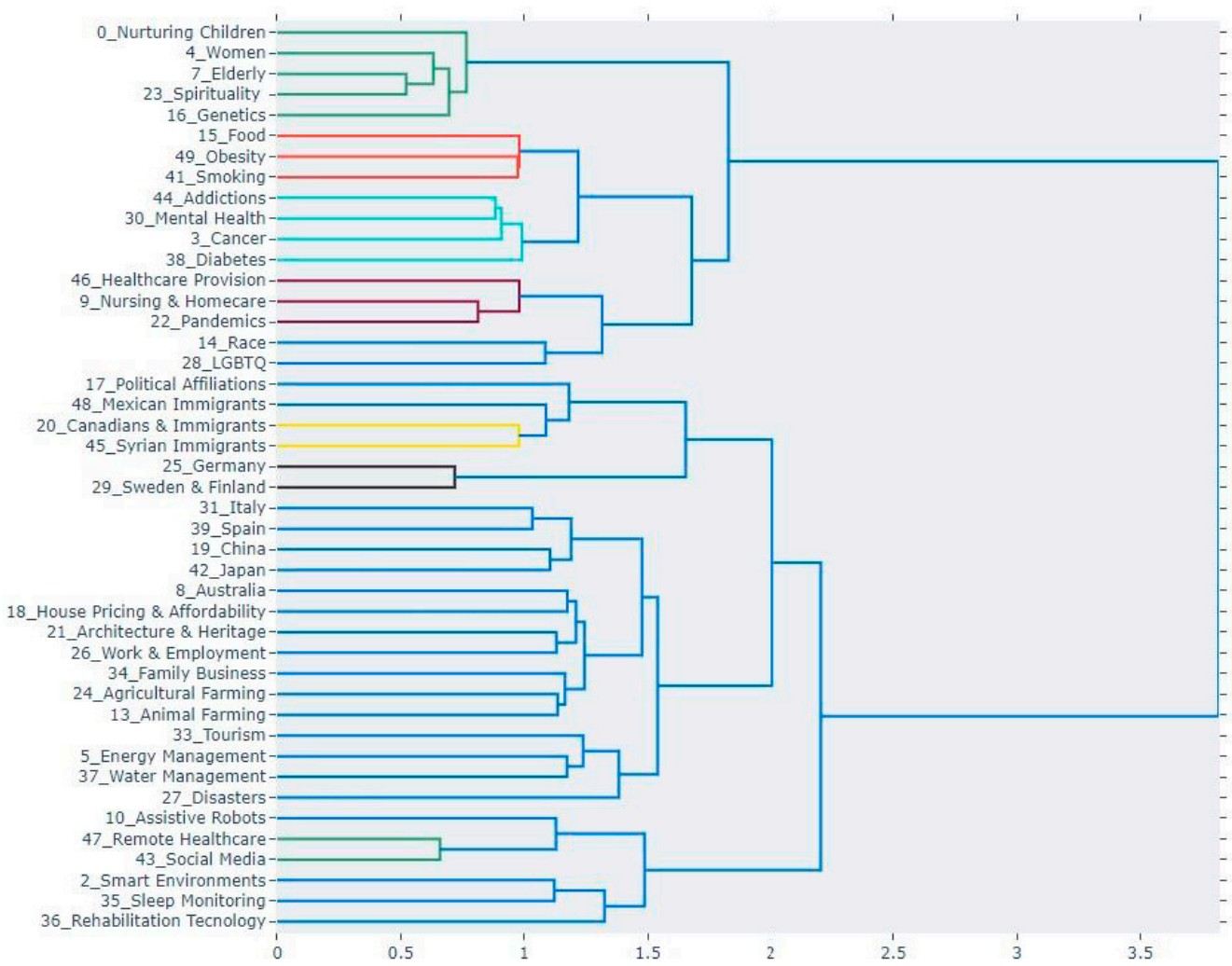

**Figure 11.** Hierarchical Clusters (Data Source: Scopus).

4.3.3. Elderly

The Elderly parameter is regarding elderly care and health. It is represented by keywords, detected by our model, such as old, elderly, old adult, home, live, adult, old people, health, life, social, elderly people, fall, daily, study, support, sensor, caregiver, aged, home care, and base. This parameter captured some concerns related to elderly people's lifestyle, health, psychological conditions at care homes, and assistive technologies. For example, Cook et al. [86] noticed that moving to a care home for the first time compromises an older person's sense of security that comes from their own 'home'. After some time, they have developed a sense of belonging to a care home as "home" and prefer 'living with care' rather than 'existing in care'. Henkel et al. [87] emphasized the importance of connecting with others and one's own personal past. They considered this valuable for many older adults living in long-term care facilities as it serves different psychosocial functions in various settings to reflect on the past and share recollections with others. On the other hand, Roh et al. [88] discovered that seniors living with families in South Korea had significantly higher life satisfaction than those living alone. Further, individuals who maintained both their social and physical activities and their economic profile showed higher levels of life satisfaction than their counterparts.

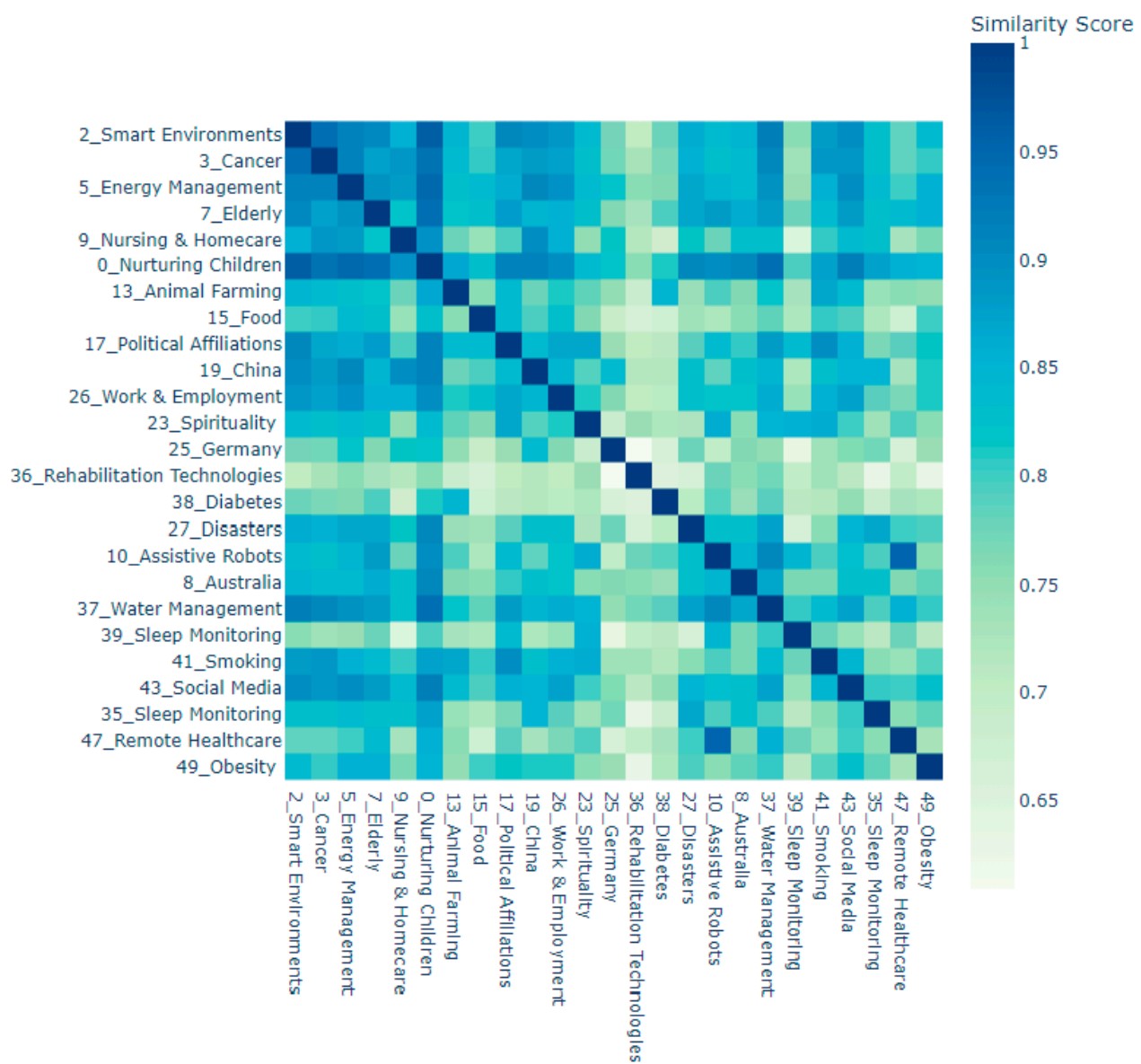

**Figure 12.** Similarity matrix (Data Source: Scopus).

### 4.3.4. Spirituality

The Spirituality parameter captures various aspects of the family spirituality including religion, faith, religious education, and religion-related social studies. It is represented by the following keywords: religious, religion, church, spirituality, Christian, chaplain, spiritual care, Muslim, prayer, faith, theology, community, study, protestant, theological, catholic, law, secular, patient, and marriage. Some examples articles from this parameter are discussed as follows. Thanissaro [89] studied and evaluated the attitude towards Buddhism in adolescents and teenagers in the UK. The involvement in religiosity was measured through temple attendance, scripture reading, meditation, spiritual experiences, and religious style. Peri-Rotem [90] discussed religion and fertility and tried to understand the changing relationships between religion, childbearing, and family formation patterns in Britain, France and the Netherlands.

### 4.3.5. Summary and Temporal Analysis

Table 2 provides a summary of all parameters in this macro-parameter. The temporal progression of the Nurturing Families macro-parameter is plotted in Figure 13. It includes four parameters: Nurturing Children, Women, Elderly, and Spirituality. We can see that spirituality is less important than other issues. The highest issues it appears is Nurturing Children followed by Woman, Elderly, and Spirituality. Nurturing Children had a peak value around 1400 in 2021.

**Table 2.** Macro-Parameter Summary (Nurturing Families).

| Parameter | Description | Sample Works |
|---|---|---|
| Nurturing Children | Focuses on the development of children along with the involvement of parents and teachers. | [70,71,76,80,82] |
| Woman | Covers different roles women play in the families & homes, challenges, issues, advice, and technological enablers while developing, maintaining, and taking care of their families. | [84,91–94] |
| Elderly | Captured some concerns related to elderly people's lifestyle, health, psychological conditions at care homes, and assistive technologies. | [86–88,95] |
| Spirituality | Captures various aspects of the family spirituality including religion, faith, religious education, and religion-related social studies. | [96–98] |

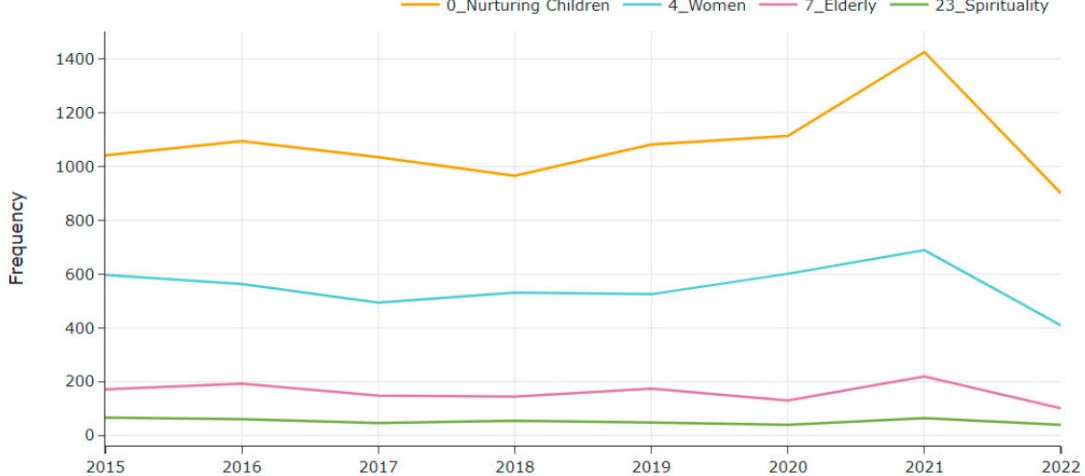

**Figure 13.** Temporal Progression (Macro-Parameter: Nurturing Families) (Data Source: Scopus).

### 4.4. Health & Lifestyle

This macro-parameter concerns health and lifestyle touching upon lifestyles, diseases and provision of healthcare. It captures the following parameters: Genetic, Food, Obesity, Smoking, Addictions, Mental Health, Cancer, Diabetes, Healthcare Provision, Nursing & Homecare, and Pandemics.

### 4.4.1. Genetics

This parameter captures the effects of genetic characteristics on families and homes. This parameter explores the following keywords mutation, genetic, gene, protein, disease, pedigree, phenotype, patient, study, exome sequence, female, clinical, nucleotide, autosomal, male, dna, genotype, protein human, polymorphism, and novel. Genetics is believed to be responsible for many developmental abnormalities. For example, NOG-related symphalangism spectrum disorder (NOG-SSD), intellectual disability and other developmental abnormalities have been reported by Pang et al. [99] as a result of microdeletions in chromosome 17q22, where the NOG gene resides [100]. Similarly, Toyoda et al. [99] studied a case of childhood-onset hyperuricemia and early-onset gout caused by dysfunctional

ABCG2. Kertesz et al. [101] studied the relationship between proximal genetic abnormality and pathogenic pathways in families.

### 4.4.2. Food

This parameter captures issues related to the family's and home's food consumption and eating habits. The keywords that were detected by our model included food, eat, meal, nutrition, food waste, household, restaurant, study, school, consumer, diet, dietary, health, vegetable, grocery, food insecurity, nutritional, food consumption, cooking, and food security.

An analysis of food-related life satisfaction by Schnettler et al. [102] found that health, family, and eating are factors that contribute to overall life satisfaction, and family interaction that occurs while eating may play an important role in overall life satisfaction. Similarly, Jackson et al. [103] explained how family homes are considered a key setting for developing lifelong eating and physical activity habits and how family home nutrition and physical activity environments influence food insecurity and childhood obesity. Lambert et al. [104] found that dinner conversations and settings are among the best times to discuss hard issues with family and friends. For example, the topic of death could be communicated via humour over dinner conversations, and they specified six different types of humour that can be used in conversations about death: entertaining humour, gallows humour, tension-relieving humour, confused/awkward laughter, and group humour/narrative chaining.

### 4.4.3. Obesity

This parameter focuses on obesity: prevalence, causes, effects, treatments, and prevention in families including in children and adolescents. It is represented by keywords such as obesity, overweight, childhood obesity, prevalence, health, adult, age, female, physical, family, high, male, and adolescent.

Childhood obesity is highly influenced by the family and home environment, according to Knowlden et al. [105]. As the global obesity epidemic affects children today and some of these children also have other disease-related problems, such as diabetes type II, Rio et al. [106] studied the impact of a gamified educational program on patients with obesity and diabetes, resulting in positive changes to their medical records and health habits. Around the world, obesity has been on the rise among children and adolescents. Socioeconomic status plays an important role in adolescents' risk of obesity, as Ni et al. [107] stated. They investigated the role of socioeconomic status on weight gain among adolescents exposed to economic and social transitions. Mosha et al. [108] observed the prevalence of childhood overweight and obesity in urban and private schools in low- and middle-income countries. Thus, it is important to ensure that there are playgrounds at schools and to encourage children to participate in physical activities.

### 4.4.4. Smoking

The Smoking parameter regardssmoking's effects on families, homes, health, culture, and environment. It is represented by keywords (detected by our model) such as smoking, tobacco, cigarette, smoker, smoke free, tobacco product, smoking cessation, adolescent, quit, study, adult, tobacco use, nicotine, home, health, secondhand, shs exposure, passive smoking, female, and free.

According to Golestan et al. [109], around the world, cigarette smoking is not considered as dangerous as narcotic drugs, since there is no country in which adolescents do not smoke. However, cigarette smoking among adolescents is considered a global and complex public health problem as it affects the development of a person's physical, emotional, spiritual, and social well-being. Therefore, they believe that families and self-efficacy are crucial in preventing young children from smoking. Leite et al. [110] stressed on that the use of tobacco can cause cancer and affect the microbiota of the oral, fecal, duodenal mucosa, and bowel luminal microbiome. Mbongwe et al. [111] noticed that the effects of culture and

environment on smoking among adolescents were greater in boys than in girls and peer and social acceptance were strong predictors of smoking among boys. As an alternative to cigarettes, Patel et al. [112] referred to the electronic nicotine delivery system called JUUL for current adult smokers. It helps them to quit smoking. Furthermore, they mentioned the importance of cessation intervention efforts and policy development to assist smokers in quitting.

### 4.4.5. Addictions

This parameter is described by the following keywords: alcohol, drink, gambling, drinking, drug, overdose, opioid, adolescent, naloxone, alcohol use, consumption, cocaine, male, study, alcohol consumption, female, family, alcoholic, health, and disorder.

Factors influencing the recurrence of drug abuse include internal factors such as ineffective family roles and functions, family imbalances, economic status, and communication and external factors, including environmental and peer group variables, as well as free time [113]. Raharni et al. [113] stressed that friends played a very dominant role in causing relapse, and external factors such as social pressures and the environment was more likely to lead to relapse. Moreover, drug types also affected the likelihood of relapse. Therefore, they affirmed on importance of social support for recovering from drug addiction. According to Cordova et al. [114] eco-developmental factors, including family and community, play a significant role in alcohol and drug use. Gajewski et al. [115] noticed that a significant number of people with depression and substance abuse disorder have comorbid substance abuse and addiction and they are more likely to have a worse quality of life and commit suicide. A link between alcohol use and anxiety issues among women with HIV was observed by Ge et al. [116].

### 4.4.6. Mental Health

People's and family mental health issues are captured by this parameter. The following keywords were detected by our model: dementia, caregiver, people dementia, live, alzheimer, dementia care, live dementia, home, dementia family, family, patient, person dementia, disease, health, family caregiver, aged, nursing, staff, and Alzheimer's disease cognitive. This parameter discovered research focused on supporting and dealing with family members that suffer from mental conditions and their caregivers including getting family help, care at the organizational level, establishing trust with caregivers, everyday conversation, and assistive technologies. For example, Berry et al. [117] noticed that people with dementia seemed to be aware when their condition was declining and frequently asked their families for help. As a result, families need to consider the functional abilities of demential elders and manage their involvement in activities. Person-centered care at the organizational level is believed to be a possible supporting way for the quality of life of people with dementia [118]. As people with dementia tend to have a decline in their ability to engage in everyday conversation, Sluis et al. examined the use of conversational analysis to improve communication between people with dementia in residential care and their caregivers [119].

### 4.4.7. Cancer

The Cancer parameter is regarding cancer patients of various ages, different cancer types, social support, and detection. It is represented by keywords (detected by our model) such as protein, cell, cancer, gene, mouse, tumour, animal, receptor, kinase, molecular, drug, enzyme, family, cell line, breast, genetic, domain, gene expression, acid, and DNA. Looking at the Scopus academic articles that belong to this parameter we were able to find a number of topics related to it. For example, Gage-Bouchard et al. [120] discussed different types of support the parents of children with cancer receive from a broad range of people (e.g., friends, family, neighbours, and health care professionals) including emotional support and information about health issues. Huang et al. [121] stressed the role of family communication patterns, coping, and well-being as they found that it is essential to support

and help cancer patients including breast, prostate, and mixed cancer types. Regarding breast cancer, as a significant health issue among women and early detection makes treatment easier and more effective, medical imaging has been used in a variety of ways to investigate breast cancer. Moreover, Melekoodappattu et al. [122] developed a computer-aided diagnostic (CAD) system to detect breast cancer by interpreting mammograms and identifying tumours with 99.33 % accuracy.

### 4.4.8. Diabetes

The Diabetes parameter is represented by keywords including diabetes, type diabetes, glucose, insulin, patient, blood, diabetic, study, health, diabete, insulin dependent, disease, dependent diabetes, adult, risk factor, family history, female, non-insulin, male, and history. This parameter covers research related to diabetes such as self-care, family history, and medications. For example, Baig et al. [123] stressed on the importance of self-care in managing diabetes in adults and discovered that participation of family members in diabetes self-care has a positive impact on patient outcomes. Among the Chinese, Zhang et al. [124] investigated the possible association between family history risk and the prevalence of diabetes. Lan et al. [125] explained how family history of diabetes and hypertension were both significantly associated with an increased risk of metabolic syndrome (MS).

### 4.4.9. Healthcare Provision

The Healthcare Provision parameter captures various dimensions of improving healthcare provisioning and developing healthcare professionals. The parameter's keywords include student, medical, nursing, medical student, education, nursing student, clinical, undergraduate, patient, medical education, health, nurse, medical school, medicine, education medical, practice, nursing education, study education nursing, and interprofessional. The dimensions covered by this parameter include the challenges of medical education including cost [126], distance learning [127] and practical training [128], how to improve family medicine [129], how to improve homecare and the nursing profession [130], how to improve medical education and profession [127], and simulations and other methods based on the emerging technologies to improve healthcare [131].

### 4.4.10. Nursing & Homecare

This parameter is about homecare where the services by healthcare professionals are made available to people at their homes as opposed to hospitals, nursing homes or elderly homes. These services could be tailored for people, elderly or young, who are ill, disabled, or waiting for their death due to their various diseases and health conditions. The parameter touches on various issues, diseases and stakeholders including patients, nursing and nurses, death, home, aged, clinical, healthcare, female, disease, medication, tuberculosis, physician, nursing home, and family. Examples of academic research under this parameter include ensuring the quality of life for the elderly through both physical and psychosocial care in nursing homes by offering a home-like environment, communication, conversations, and self-care, which are all considered high marks of a quality elder care program [132], ethics of families in healthcare decision making [133], the value of death conversations in the clinical setting and how conversations about death early on in life can be an effective way for doctors and patients to co-create more healthy lifestyles [134], the end-of-life care policy and how patients are encouraged to die at home, which most believe to be their preferred setting [135], and how in-home care using the remote monitoring and diagnostic system would provide a holistic view of the patient's health status and minimizes hospitalization times [136].

### 4.4.11. Pandemics

This parameter captures the effects of Pandemics on families and homes across the world. The various dimensions captured include vaccine, infection, virus, influenza, viral epidemic, spread, respiratory, outbreak, infant, patient, pneumonia, and social aspects. Examples of studies under this parameter include how family resources during a pandemic are re-allocated as a result of a health shock to one child [137], the use of vaccination and social distancing measures to slow the spread of infectious diseases and the provided public support to enhance epidemic control [138], the behavioral measures (e.g., wearing face masks and keeping social distance from others) for combating COVID-19 in the absence of vaccines or causal therapies [139], how family violence during COVID-19 has led to a lock-down and long-term home isolation across China [140], early detection frameworks using smartphone sensors on individual smartphones globally to identifying the disease early, monitoring the users' current location, and taking protective measures [141], opportunities and benefits opened to those in need within society, the community, and the nation such as changes undergone to healthcare including teleconsultation, Internet use, e-learning, and the effect on water sources, waste management systems, and conservation of natural resources [142].

### 4.4.12. Summary and Temporal Analysis

Table 3 provides a summary of all parameters in this macro-parameter. The temporal progression of the macro-parameter Health & Lifestyles which includes 11 parameters is shown in Figure 14. Among the other parameters, cancer parameter was highly discussed in 2021 and had the highest peak value of more than 800. We note that in 2019 as the time of spreading COVID-19 pandemic might affect in the number of research studies of some parameters like Nursing & Homecare, Food, Pandemics.

**Table 3.** Macro-Parameter Summary (Health & Lifestyle).

| Parameter | Description | Sample Works |
|---|---|---|
| Genetics | Captures the effects of genetic characteristics on families and homes. | [99–101,143] |
| Food | Captures issues related to the family's and home's food consumption and eating habits. | [102–104,144,145] |
| Obesity | Focuses on obesity prevalence, causes, effects, treatments, and prevention in families including children and adolescents. | [105–108] |
| Smoking | Regarding smoking's effects on families, homes, health, culture, and environment. | [109–112,146] |
| Addictions | Covers how family and community play a significant role in drug abuse and recurrence. | [113–116,147] |
| Mental Health | Captures People's and family mental health issues. | [117–119,148] |
| Cancer | Describes cancer patients of different ages, cancer types, supportive care, and cancer detection. | [120–122,149] |
| Diabetes | Covers research related to diabetes such as self-care, family history, and medications. | [123–125,150] |
| Healthcare Provisioning | captures various dimensions of improving healthcare provisioning and developing healthcare professionals. | [126–131] |
| Nursing & Homecare | Homecare is a form of healthcare where people receive healthcare services at home rather than in hospitals. | [132–134,136] |
| Pandemics | A global illustration of how pandemics affect homes and families. | [137–142] |

### 4.5. Communities & Nations

This macro-parameter covers various dimensions of families and homes distinguished by different communities, countries, societies, or nations. It captures the following parameters: Race, LGBTQ, Political Affiliations, Mexican Immigrants, Canadian and Immigrants, Syrian Immigrants, Germany, Sweden and Finland, Italy, Spain, China, Japan, and Australia.

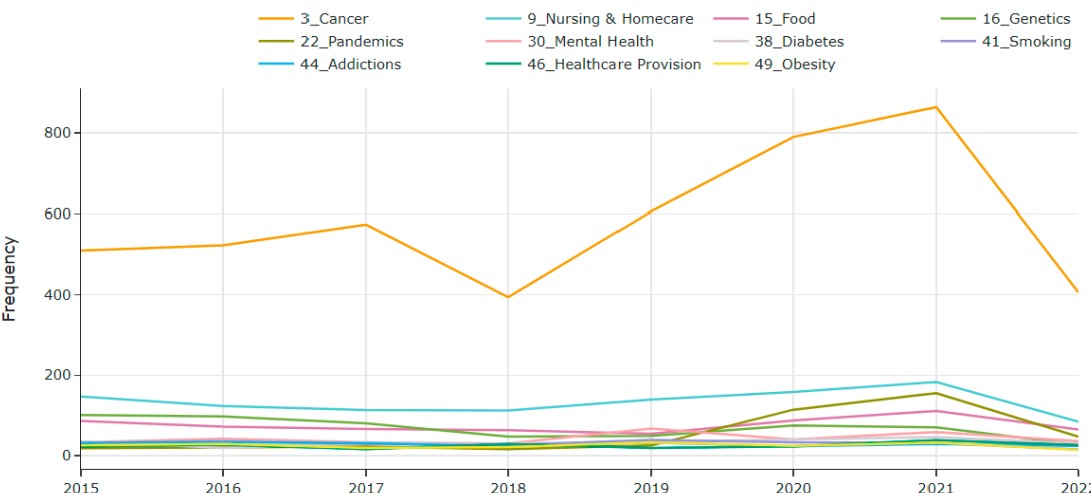

**Figure 14.** Temporal Progression (Macro-Parameter: Health & Lifestyles) (Data Source: Scopus).

4.5.1. Race

The Race parameter captures the racial issues associated with families and homes. It includes the following keywords: black, racial, race, african, racism, family, health, hispanic, social, black woman, american, woman, slavery, segregation, child, black, white, african american, racial ethnic, black family, and research. The parameter discloses several important dimensions in academic literature including the person's identity and how it can be affected by the families and places they visit [151,152], how freedom, justice, and equality are constrained by social and political conditions [153], family, friends, or members of the same racial group reporting racial discrimination incidents [154], racialized discrimination against multiracial families [155], racial socialization and how parents transmit race-related messages about race and racism to their children through racial socialization [156], the online genetic-genealogical networks and genetic relative match role in the reconstruction of historical and genetic connections between white and black families which could break racial barriers and help uncover the family history knowledge of slavery [157], and the school experience impact on blacks and their families [158].

4.5.2. LGBTQ

The lesbian, gay, bisexual, transgender, and questioning (LGBTQ) parameter captures the families and home issues associated with these distinct groups. The keywords that were detected by our model include gay, transgender, bisexual, sex, male, heterosexual, lesbian, homosexuality, parent, adult, health, gay bisexual, gay man, social, study, queer, sexual gender, female, youth, and father. Academic documents under this parameter discussed various issues related to LGBTQ including the behaviour of lesbian, gay, bisexual, and transgender couples alters gender roles in the home [159], the family role in the construction of a person's sexual identity in terms of acceptance and integration into familial roles [160], the public health concern of sexually transmitted diseases (STDs) prevalence among heterosexual males [161], health and social care provision concerns related to LGBTQ individuals [162], and transgender youth and medical fertility preservation to avoid infertility or biological sterility [163].

4.5.3. Political Affiliations

This parameter shows how the political candidate's identity is framed and which information is most valuable to voters in order to vote for him or her. The effects of socio-political change on family relationships, especially for LGBTQ individuals. As well as a political perspective on COVID-19. Our BERT model captured some keywords related to this parameter as party, election, political, vote, electoral, voter, presidential, candidate, house, partisan, president, parliamentary, voting, parliament, democracy, congressional,

democratic, trump, legislator, and political party. Sclafani [164] explained political identities can be framed using inherited identities derived from family. Rugh [165] explored how migration, race, and class affect Latino household wealth.

### 4.5.4. Mexican Immigrants

This parameter covers issues related to families and homes that are specific to Mexico, Mexican people, and Mexican immigrants in the US. The keywords in this parameter include immigrant, parent, adolescent, migrant, migration, study, mother, familism, health, mexican immigrant, female, experience, social, transnational, human, adult, policy, and mexican american. For example, Fuller-Iglesias [166] believes that migration affects family members, especially the wives and children of migrants. Tsai et al. [167] explained the relationship between parental stress and adolescents' emotional support for family members. On days when parents were experiencing family stressors, adolescents were more likely to offer support to other family members than to their parents. For Mexican family constellations therapy, Duncan [168] claimed that non-native therapeutic practice may align with local cultural frameworks to promote new forms of therapeutic engagement and social interaction.

### 4.5.5. Canadian & Immigrants

This parameter includes keywords, Canadian, family, study, child, social, indigenous, policy, community, health care, research, home care, practice, mental, adult, article, life, patient, physician, mental health, and immigrant. The documents under this parameter captured some issues related to Canadian people & immigrants in general such as Schieman et al. [169] studied the influence of work-family multitasking on Canadian and American Workers and identified four main factors: Social status, employment type, work location, and schedule, and job demands and resources. Laplante [170] explained that families in Canada are at risk due to changes in values, gender relations, and the legal framework dealing with family life. These changes still affect many families today. As a result, Canadian society is moving toward immigration. Burrage et al. [171] focused on trauma healing in the Indian Residential School system of Canada and found that a strong sense of belonging to family, culture, and community is critical to helping and healing after being physically, psychologically, or sexually abused.

### 4.5.6. Syrian Immigrants

This parameter explores the following keywords refugee, syrian, immigration, deportation, asylum, syrian refugee, migration, migrant, family, country, resettlement, displacement, camp, child, detention, mental health, home, asylum seeker, interview, and article. It captures issues related to Syrian people and immigrants such as how the home is associated with feelings of belonging and meaningful relationships [172], how the material and symbolic conditions of living in exile reshape the concept of home in long-term [172], how the refugee camps can be as "homes" in view of their persistence and resilience [173], risks and limitations children and their parents can be subject to due to undocumented immigration status [174], the concept of Multigenerational Punishment and how strong social ties, constant interaction, and connections make it common among families [174], trauma exposure and post-traumatic stress symptoms among Syrian youth refugee and the need for school- or family-based intervention to address this issue, in particular within a Syrian Muslim cultural context, to distinguish between the needs of girls and boys [175], psychosocial and environmental stressors as well as perceived barriers to health care the parents of Syrian refugee children face [176], and harm of the loss of educational opportunities causes by this kind of crisis to adolescents and its long-term devastating outcomes [177].

### 4.5.7. Germany

This parameter is represented by keywords including german, family, war, home, study, article, social, life, history, germany, child, work, research, author, migration, country, century, health, refugee, and memory. Several dimensions are captured by our model related to Germany including post-war issues, childlessness and birthrate decline, educational opportunities, leaving and returning to parental homes, and identity development and international mobility challenges in adolescence. For example, Mouton [178] discussed the great struggle to provide for the missing, lost, and displaced children who suffered in post-war Germany and how successful the efforts to care for these children were [178]. Hill [179] believes that childlessness has become a reality for more women in Germany as the birthrate declines. Consiglio et al. [180] observed that even though equal opportunities are a priority for German policymakers regardless of an individual's socioeconomic background, children from families without academic backgrounds continue to be disadvantaged in educational opportunities.

### 4.5.8. Sweden & Finland

This parameter covers concepts about home, family, child, parental, language, life, education, social, and health related to Sweden and Finland. Academic research under this parameter discussed various aspects including home construction, family theory, parental relation with their children, etc. For example, Yakovleva et al. [181] stated that in family theory, in the Finnish paremiologically, there are three main levels: Nuclear Family, Androcentric Family, and Affinal Family [181]. Edman [182] confirmed that culture played a significant role in constructing a home in Sweden. For the Russian–Swedish families, Abreu Fernandes [183] stressed the importance of embedding mother-child communication in mundane activities such as family activities and home language lessons. A study by Gauffin et al. [184] in Sweden found that a dysfunctional family environment can lead to an increased likelihood of adult alcohol-related illnesses for children. Furthermore, growing up in a low socioeconomic position household will increase a child's vulnerability, including low school performance, low education levels, poorly paid, stressful jobs, occupational hazards, and poor general health. Several studies under this parameter focused on family engagement with their children, especially for Finnish Parents.

### 4.5.9. Italy

The families and homes research related to Italy have focussed on dimensions including psychopathology of children, sociocultural and socio-political factors, factors for employment prospects, the significance of house architectures for Italians, and others. The parameter includes the following keywords, italian, family, social, study, work, child, migrant, language, article, history, parent, home, migration, house, immigrant, lockdown, gender, economic, and life. Examples of research captured under this parameter are as follows: Zanfi et al. [185] noticed that Italian families place great significance on their family house and its relationships with peculiar welfare models and families consider their houses to be their sole inheritance. Pepe et al. [186] found that family dysfunction might cause Italian children with psychosomatic conditions. Curdt-Christiansen et al. [187] studied the difficulty for transnational families to develop literacy in the home language for their children due to socio-cultural and socio-political realities [187].

### 4.5.10. Spain

This parameter covers concepts about spanish language, bilingual, family, vocabulary, spanish speak, study, heritage, speaker, literacy, home, parent, home language, proficiency, linguistic, read, social, and learner. For example, Abchi et al. [188] found that as Spanish heritage speakers grow up in multilingual environments, their syntactic complexity in written narratives does not differ greatly from that of full Spanish speakers. By studying two old stone-based houses in the Philippines that were resettlement by the Spanish colonists, Barretto-Tesoro [189] concluded that both the artefacts and the architectural style of the

houses indicate when it was built. Blair and Lease [190] mentioned that a combination of linguistic and social factors can contribute to differences in pronunciation of voice segments over generations of Spanish heritage speakers.

### 4.5.11. China

This parameter includes keywords, Chinese, family, child, study, social, housing, chinese family, home, market, parent, policy, culture, immigrant, migrant, filial, firm, government, economic, relationship, and research. Researchers studied various topics related to China and Chinese people, in the following we give examples of research captured by our model under this parameter. As China is a state with a father-figure leader who represents the country as a family, Steinmüller [191] believes this clearly illustrates the relationship and feeling between the leader and the country. The meaning of housing in China, especially for young people, and its role in society as a whole, as explained by Xiaoming [192], is demonstrated by its relationship to the subsystems: state politics, its culture, market economics, and urbanized at-home lifestyles. These four aspects serve as a reflection of Chinese society today.

### 4.5.12. Japan

This parameter covers issues related to families and homes that are specific to Japan and Japanese society. The issues in this parameter include care, language, family, home, child, work, caregiver, culture, life, health, medical, student, old, house, adult, and social. For example, Li [193] explains that a stable and safe home remains one of humankind's most fundamental needs in Japanese Americans' literature. Gould [194] studied the placement of Japanese Buddhist altars in their western homes and how that emphasis on that the home is a place to practice the domestic religious sphere by decorating it with concrete artefacts. Moreover, Uriu et al. [195] explained how artefacts can be used to support everyday domestic rituals of remembrance and memorialization. Ono [196] found that in rural Japan, cultural values of landscape visibility are major factors that contribute to its resident's daily life [196]. Choe [197] pointed out to a translator that can refer to the unfamiliar emotional language used in the original text also such topics as child murder and mental illness. Nakamura [198] explained that strong parental beliefs about the importance of language development led them to insist on raising bilingual children who speak a specific language and also practice their home literacy activities regularly.

### 4.5.13. Australia

This parameter explores the following keywords australian, australia, homelessness, housing, homeless, home, health, social, study, parent, young, research, mental, community, aboriginal, work, policy, article, school, and interview. This is the largest in terms of the number of documents and most diverse parameter in the Communities & Nations macro-parameter and this may be indicative of higher research activity in Australia about socioeconomic issues of its own people. For example, Ou et al. [199] found that in Australia family's socioeconomic status, health insurance coverage, and region of residence strongly influence the use of general practice services. Furthermore, Twomey et al. [200] claim that socio-economic class intersects with gender and ethnicity in structuring both the ability to pursue further education and the experiences that follow. Rowan et al. [201] found that Australian teacher preparation and education is effective at preparing teachers for many different fields of work that interfere with their ability to teach students from diverse cultural, linguistic, and socioeconomic backgrounds.

### 4.5.14. Summary and Temporal Analysis

Table 4 provides a summary of all parameters in this macro-parameter. Figure 15 plots temporal progression for 13 parameters which formulate the macro-parameters Communities & Nations. According to the figure, the Australia parameter frequency is the highest activity compared to the other parameters, and more research has been done on it. Followed by Germany, Political Affiliations, China, and Canada.

**Table 4.** Macro-Parameter Summary (Communities & Nations).

| Parameter | Description | Sample Works |
|---|---|---|
| Race | Captures the racial issues associated with families and homes. | [151,152,154–156] |
| LGBTQ | Captures the families and home issues associated with these distinct groups: The lesbian, Gay, Bisexual, Transgender, and Questioning. | [159–162] |
| Political Affiliations | Demonstrates how a political candidate's identity is framed and what information is most valuable to voters. | [164,165,202–205] |
| Mexican Immigrants | Covers issues related to families and homes that are specific to Mexico, Mexican people, and Mexican immigrants in the US. | [166–168] |
| Canadian & Immigrants | Captured some issues related to Canadian people & immigrants. | [169–171,206,207] |
| Syrian Immigrants | Captures issues related to Syrian people and immigrants. | [172–174] |
| Germany | Describes several dimensions related to Germany parental homes, adolescence, educational opportunities, identity development. | [178–180,208,209] |
| Sweden & Finland | A basic overview of the Swedish and Finnish home, family, child, parental, language, life, education, and health. | [181–183] |
| Italy | Several aspects of Italian families and homes have been discussed, including psychopathology, sociocultural and sociopolitical factors and factors related to employment prospects. | [185–187,210,211] |
| Spain | Covers concepts about Spanish home, family, social, and language. | [188–190,212] |
| China | Covers concepts about Chinese home, family, social, culture, and immigrant. | [191,192,213,214] |
| Japan | Covers issues related to families and homes that are specific to Japan and Japanese society. | [193–198] |
| Australia | It focuses on issues specific to the Australian society and families. | [199–201,215–218] |

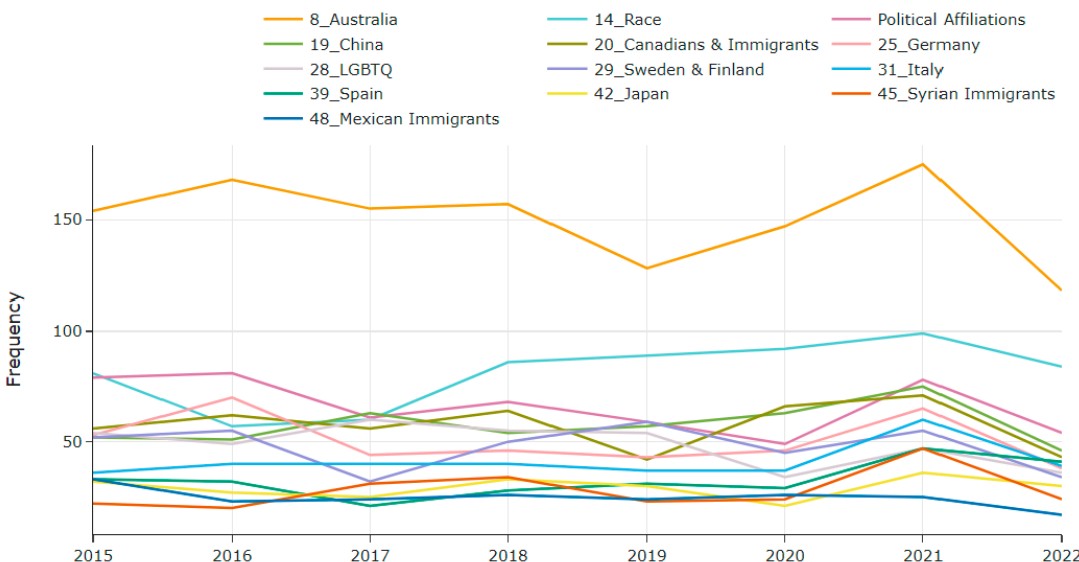

**Figure 15.** Temporal Progression (Macro-Parameter: Communities & Nations) (Data Source: Scopus).

*4.6. Resources & Management*

The macro-parameter Resources & Management concerns about establishing home resources and managing families and homes. It discusses aspects related to families and homes economics and operations. It comprises eight parameters House Pricing & Affordability, Architecture & Heritage, Work & Employment, Family Business, Agricultural Farming, Animal Farming, Tourism, Energy Management, Water Management, and Disasters.

4.6.1. House Pricing & Affordability

The House Pricing & Affordability parameter is regarding estate prices and different factors that affect the pricing and family affordability. The keywords that were detected by our model include house price, housing, house, market, real estate, estate, housing price, model, housing market, hedonic, prediction, urban, price prediction, mortgage, datum, residential, land, income, and buyer.

An exploration of housing prices in Malaysia was conducted by Osmadi et al. [219], revealing that they are affected by a multitude of factors, including structural, neighbourhood, and locational factors. Property prices must be monitored and controlled by the government. Li et al. [220] investigated macroeconomic parameters that may affect real estate prices and designed a prediction model for price fluctuation. The energy effects on house pricing were investigated by Cespedes-Lopez et al. [221]. They confirmed that qualifications for energy have an effect on the asking price of housing. Moreover, Shi [222] stated that assessments of residential real estate are important to financial institutions and municipalities that depend on property taxes as their primary source of revenue. A study to discover the nature of the relationship between housing type and prices across different regions in the UK is conducted by Hudson et al. [223].

4.6.2. Architecture & Heritage

The Architecture & Heritage parameter captures various dimensions of house design including architectural design [224], archaeology of houses [225], antiquities [226], sites and locations [227], materials and techniques [228,229], and some specific heritage homes [230]. This parameter mainly covers articles that discuss the mentioned aspects of house design rather than the relation between the house architecture and their relationship with and impacts on families. We could have removed this parameter from the analysis, however, we decided to keep it with the intention that in the future this topic of the relationship between families and house architecture and heritage can be explored.

4.6.3. Work & Employment

The Work & Employment parameter covers topics related to jobs, work environment, and work-family conflicts. Keywords under this parameter include work, employee, work family, job, family, family conflict, worker, work home, workplace, job satisfaction, study, life, stress, relationship, resource, organizational, work life, supervisor, employment, and home.

Qiu et al. [231] stated that working-family conflict is influenced by the family's boundary characteristics, as is life satisfaction. Similarly, Schieman et al. [232] attested that work-family boundaries are being blurred due to work pressures and role-blending activities. Alacovska [233] focused on creative workers and followed the 'ethics of care' perspective in exploring, acknowledging and valuing the relational, communal, moral, interpersonal, and interdependent aspects of creative work. On the other hand, Liu et al. [234] believe that educational level, job training, and years of experience are not enough to measure effective work behaviour. It is more efficient to connect individual performance to the goals of the company. Since telework from home becomes increasingly important, especially for software engineers, and by comparing office work before and after the pandemic, Smite et al. [235] found that it is possible to determine the differences in productivity between the two periods [235].

### 4.6.4. Family Business

The Family Business parameter contains keywords business, family firm, family, family business, corporate, company, governance, non-family, social responsibility, corporate governance, corporate social, family ownership, ceo, study, social, financial, sustainability, firm performance, relationship, and innovation. For example, Sanchez-Famoso et al. [236] explained that firm performance is affected by both family and non-family social capital and the non-family social capital affects firm performance more than family social capital, and it also acts as a mediator between the two. Bandelj et al. [237] clarified how the economy and the family are intertwined, whereby economists and financiers teach the logic of the instrumental market to family businesses. Across family-owned enterprises, there are multiple dimensions of "family support" from psychological and resource perspectives. Among these Munagapati et al. [238] defined five dimensions: developing a vocational identity, role modelling, providing resources, supporting human capital, and social capital. Giannakopoulou et al. [239] stressed the importance of corporate governance in determining operational and financial performance [239]. Goel et al. [240] developed a conceptual model linking family functioning to Human Resources (HR) flexibility and subsequent HR outcomes in family businesses.

### 4.6.5. Agricultural Farming

The Agricultural Farming parameter is about families involved in agriculture and farming and the various dimensions related to it such as establishing family farming [241], contemporary agricultural practice [242], agricultural recovery [243], modernizing agriculture [244], and agriculture as a main source of income, employment, and economic activity in rural areas [117]. The keywords that were detected by our model include farms, agriculture, land, family farm, farming, food, crops, households, markets, labor, sustainability, and climate.

### 4.6.6. Animal Farming

This parameter captures studies related to animal farms. It is represented by keywords including dog, home range, range, habitat, pet, chicken, cat, poultry, bird species, pig, home, wild, female, male, meat, owner, article, house, and animal. Looking at the academic documents that belong to this parameter we were able to identify some aspects of this parameter. These include human subsistence behaviour and family consumption [245], biosecurity measures in rearing practices [246], intensive poultry farming [247], factors of positive effects of dog ownership in families [248], and understanding animals' role in ecosystems [249].

### 4.6.7. Tourism

The Tourism parameter is about relations, effects, opportunities, issues, and challenges of tourism with home and family. Our model detected the following keywords for the parameter: tourism, tourist, hotel, travel, guest, holiday, tourist destination, family, sustainable, airbnb, research, cruise, accommodation, social, second home, local, house, tourism industry, heritage, business. Examples of topics captured by this parameter: kinship tourism and the opportunities it can provide for forming a family [250], the home-sharing mobile platforms such as the Airbnb app which connects tourists and local hosts [251], tourism commercial homes and their important immaterial dimensions in the point of view of Lifestyle entrepreneurs [252], how rural tourism opens up new economic opportunities for local municipalities and counties [253], and personal life, family life, home and adequate housing rights preserved by international human rights law [254].

### 4.6.8. Energy Management

The Energy Management parameter contains keywords energy, grid, electric, management, electricity, building, solar, energy management, air, appliance, renewable, residential, house, home energy, thermal, battery, indoor, heating, renewable energy, and energy con-

sumption. This parameter capture several important energy-related dimensions including energy usage patterns and various smart home systems that can improve energy consumption such as smart plugs, smart circuit breakers [255], energy efficiency and the renewable energy systems in energy-positive houses [256], decarbonization of housing for mitigating climate change and rising fuel prices [257], and Smart Grids utilities that collect and analyze consumption's data, and efficiently manage household appliances through advanced (smart) meters [258,259].

### 4.6.9. Water Management

The Water Management parameter discloses several important dimensions of water resource management. It is represented by keywords (detected by our model) such as water, water supply, drinking water, sanitation, water quality, household, drinking, supply, water consumption, urban, water treatment, drink water, rainwater, water source, water use, water management, sensor, wash, household water, and toilet. Looking at the documents that belong to this parameter we were able to capture various dimensions of this parameter including environmental sanitation and public awareness [260], water scarcity and insecurity including required interventions [261], polluted water sources and rainwater harvesting [262], water conservation [263], and water consumption prediction models [264].

### 4.6.10. Disasters

This parameter captures the effects of floods, earthquakes, and other disasters on families and homes. The keywords include flood, disaster, earthquake, hurricane, evacuation, coastal, landslide, community, tsunami, flooding, flood risk, impact, event, natural disaster, recovery, household, storm, housing, and resilience. The overarching theme of the documents in this parameter is the disaster management and decision-making process. For example, Dobson et al. [265] observed the influence of visualizing flood hazards on house purchase decisions and Faulkner et al. [266] claimed that the number of flood-affected homes has decreased due to flood frequency analysis and risk management. To help people recover from trauma such as earthquakes, Warner et al. [267] considered self-efficacy and social support to be coping resources. Chatterjee et al. [268] found that COVID-19 lockdowns reduce noise levels, making low-magnitude earthquakes easier to detect. As natural disasters have become more frequent across the world, Jauhola [269] noticed that the paradigm of "build back better" is emerging as a way to encourage resilience and long-term development.

### 4.6.11. Summary and Temporal Analysis

Table 5 provides a summary of all parameters in this macro-parameter. The temporal progression of the macro-parameter Resources & Management, which includes tenth parameters, is shown in Figure 16. We observed that all parameters have similar activities except the Energy Management parameter which has the highest activity referring to its important. For the parameters Agricultural Farming and Animal Farming, there is a few papers that has publication date on 2023.

### 4.7. Technologies

The Technologies macro-parameter captures parameters related to the utilization of technology in various aspects related to the families, homes, and individuals to improve the quality of life. It captures the following parameters: Assistive Robots, Remote Healthcare, Social Media, Smart Environments, Sleep Monitoring, and Rehabilitation Technologies.

**Table 5.** Macro-Parameter Summary (Resources & Management).

| Parameter | Description | Sample Works |
|---|---|---|
| House Pricing & Affordability | It is about estate prices and how they are influenced by different factors. | [219–221] |
| Architecture & Heritage | Describes a wide range of house design dimensions. | [224–230] |
| Work & Employment | Work-family conflicts, the work environment, and jobs are covered. | [231–234] |
| Family Business | Provides information about family businesses, ownership, performance, relationships, and innovation. | [236–240] |
| Agricultural Farming | Concerns related to farming and agriculture families and the varying dimensions involved. | [245–249] |
| Tourism | Describes the relations, effects, opportunities, issues, and challenges of tourism with home and family. | [250–253] |
| Energy Management | Captures several important energy-related dimensions including energy usage patterns and various smart home systems that can improve energy consumption | [255,257–259] |
| Water Management | Discloses several important dimensions of water resource management. | [260–263] |
| Disasters | Captures the effects of floods, earthquakes, and other disasters on families and homes. | [265–269] |

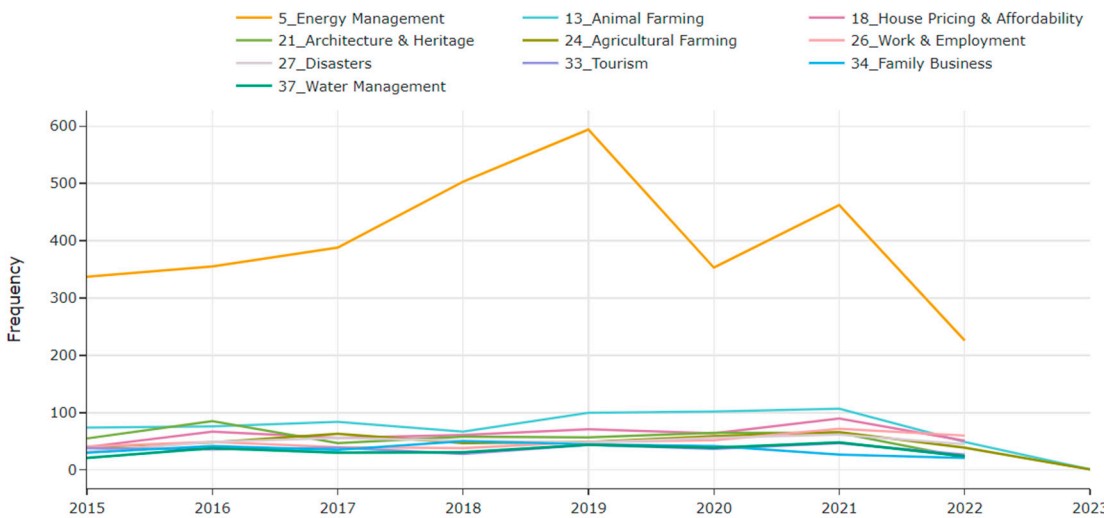

**Figure 16.** Temporal Progression (Macro-Parameter: Resources & Management) (Data Source: Scopus).

### 4.7.1. Assistive Robots

The Assistive Robots parameter is represented by keywords including robot, robotic, home, human robot, robot interaction, design, service robot, environment, mobile robot, user, mobile, social robot, assistive, base, social, machine, propose, intelligent, paper, and technology.

Parisi et al. [270] believe that assistive mobile robots have the capability of operating in complex environments such as houses, which are ideal for self-care and independent living applications and enhance the perception of safety among elderly people and prevent them from losing their confidence in their homes. For example, a home biomonitoring robot can sense and recognize human activity in a living environment by using active sensing [271], Humanoid robots can interact with the elderly in nursing homes using spoken dialogue which provides companionship and helps the elderly stay mentally active [272], in the home environment, a robot can assist the elderly in finding misplaced items [273], and the smart housekeeper (an IoT-based indoor mobile robot) can provide housekeeping services and control home appliances and provide indoor security [274].

### 4.7.2. Remote Healthcare

This parameter highlights the use of technologies in developing better lifestyles and healthcare. The keywords corresponding to this parameter are listed below: health, patient, healthcare, medical, internet, datum, sensor, internet thing, monitor, remote, health monitoring, security, wearable, network, health care, home, cloud, technology, mobile, and privacy.

Bendahan et al. [275] believe that Virtual Care (VC) can make health care more accessible, more affordable, and more available though it may not be appropriate for all clinical scenarios, especially when a thorough examination is required. Jita et al. [276] explained that using IoT technology in in-home care offers a number of benefits including allowing caregivers and physicians to monitor patients remotely, empowering the patients to be more independent, reducing costs and giving loved ones peace of mind. Similarly, Huifeng et al. [277] noticed that IoT devices are used widely to reduce health-related risk factors, such as wearable sensors for a continuous health monitoring system for athletes. However, due to the vulnerability of these wearable devices' data transmissions, privacy and integrity need to be protected. Jan et al. [278] presented a lightweight and secure communication approach for data exchanged among the devices of healthcare infrastructure. Alam et al. [279] affirmed that being physically, mentally, and socially healthy can all be enhanced by a healthy living environment and introduced an ambient assisted living framework for predicting the emergency state of psychiatric.

### 4.7.3. Social Media

The Social Media parameter is represented by keywords including social medium, twitter, facebook, online, user, networking, social networking, tweet, social network, network, news, networking online, youtube, friend, online social, broadband, internet, study, digital, and video.

Looking at the documents that belong to this parameter we were able to capture various dimensions of this parameter including using social networking sites such as Facebook, one can obtain and provide social support, which is widely regarded as important for one's health [280], how social media technologies are being used to enable users to suggest policy changes and solicit support for them [281], the diversity of content generated through ICT provides a platform for a free press to inform and debate instead of being controlled by traditional media channels [282], using the analysis of social media platforms (e.g., Weibo and Twitter) we can better understand the Challenges of Working from Home (WFH) during COVID-19, such as long work hours, family and food commitments, and health concerns [283], and Twitter and social media commentaries revealed two particular trends: first is a result of computational processes that analyse and mine data, and second is a result of financial algorithms that make automated trades [284].

### 4.7.4. Smart Environments

This parameter focuses on intelligent and smart environments and the Internet of Things (IoT). It is represented by keywords including internet, network, security, internet thing, sensor, automation, wireless, base, technology, intelligent, malware, datum, detection, home automation, attack, and mobile. Looking at the academic research that belongs to this parameter we were able to find a number of topics that capture various dimensions of this parameter. These include sensing, activity recognition, smart system design, wireless network, smart cities and homes, smart home system security, energy management systems, ambient intelligent environment, and virtual assistants. For example, Rabinowitz [285] discussed a sense of home which is a brain-wide phenomenon that involves many factors, including memory, language, emotions, and cognition. A study by Benmansour et al. [286] explained how multi-occupant smart homes could recognize human activity and that identifying residents and recognizing human activities are the two main factors causing difficulty. Shih et al. [287] emphasized the importance of improving the quality of living and reducing construction costs in response to society's economic development needs.

### 4.7.5. Sleep Monitoring

The Sleep Monitoring parameter capture dimensions related to monitoring sleeping habits and discovering sleeping disorders. The keywords that were detected by our model include sleep, apnea, sleep quality, sleep apnea, sleep stage, night, polysomnography, sleep disorder, disorder, sleep duration, stage, study, home, sleep, sleep research, wake, adolescent, patient, sleep monitor, and male. Using an in-home sensor-based sleep assessment methodology, it is possible to continuously monitor sleep behaviour and other activities that may affect mental performance in healthy older adults [288]. Electroencephalograms (EEGs) can be used for sleep stage classification, which is highly desirable for many emerging technologies, including telemedicine and home healthcare [289]. Sharma et al. [290] were able to identify six sleep disorder types using electroencephalography signals, including insomnia, nocturnal frontal lobe epilepsy (NFLE), narcolepsy, rapid eye movement disorder (RBD), periodic leg movement disorder (PLM), and sleep-disordered breathing (SDB). Radha et al. [291] designed a home-based system for monitoring sleep using heart rate variability (HRV) may be a cost-efficient and ergonomic alternative to polysomnography. Honda et al. [292] explained how flexible, wearable sensors, such as a mask-borne flexible humidity sensor, can be used to measure respiratory rates during sleep to diagnose sleep apnea symptoms.

### 4.7.6. Rehabilitation Technologies

This parameter captures studies related to home-based training and rehabilitation systems. It consists of the following keywords: rehabilitation, exercise, patient, parkinson, game, stroke, training, disease, home, base, therapy, motor, virtual, home base, pd, week, feedback, wearable, and upper limb. Looking at the academic documents that belong to this parameter we were able to identify some aspects of this parameter. These include cognitive rehabilitation using computers is a promising way to update working memory in Parkinson's disease patients [293], the use of an Automated Rehabilitation System (ARS) for physical rehabilitation in rehabilitation clinics (e.g., knee and hip replacement clinics) [294], variety of rehabilitation games (e.g., the Microsoft Kinect and Nintendo Wii Balance Board) are available for range-of-motion and balance training [295], a stroke rehabilitation game includes players receive haptic feedback based on facial expressions for patients who suffer from a combination of motor and sensory dysfunction and central facial paralysis [296], and designing smart mirrors to assist people who cannot receive professional guidance in standardizing and correcting their actions [297].

### 4.7.7. Summary and Temporal Analysis

Table 6 provides a summary of all parameters in this macro-parameter. Figure 17 shows the temporal progression of the Technologies parameters. More research has been done on the Smart Environment parameter than other Technologies parameters. It appears that there is drop in 2020 in the Smart Environment research activities which might be due to the COVID-19 pandemic as more research focus on the pandemic itself.

**Table 6.** Macro-Parameter Summary (Technologies).

| Parameter | Description | Sample Works |
|---|---|---|
| Assistive Robots | Describe various robot services that can be used in the home environment and it's the benefits. | [270–273] |
| Remote Healthcare | Highlights the use of technologies in developing better lifestyles and healthcare. | [275–278] |
| Social Media | Describes a variety of aspects related to social support, technology use, and analysis of social media platforms. | [280–282] |
| Smart Environments | Focuses on intelligent and smart environments and the Internet of Things (IoT). | [285–287,298,299] |
| Sleep Monitoring | Reviews dimensions related to monitoring sleeping habits and discovering sleeping disorders. | [288–292] |
| Rehabilitation Technologies | It related to home-based training and rehabilitation systems. | [293–296] |

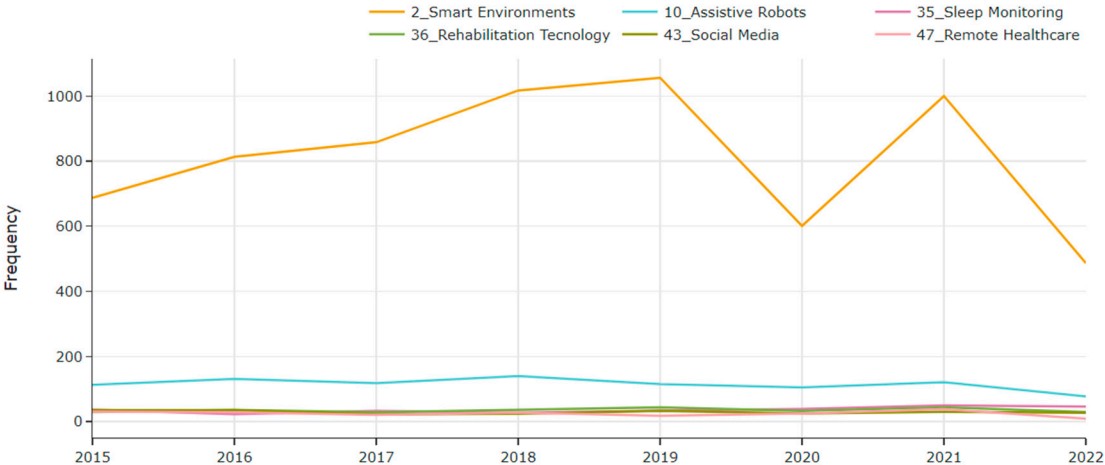

**Figure 17.** Temporal Progression (Macro-Parameter: Technologies) (Data Source: Scopus).

## 5. Parameter Discovery for Families & Homes (Public: Twitter)

The purpose of this section is to discuss the parameters that our BERT model detected from Arabic Twitter dataset which includes 930,110 tweets. it reflects the view of public in Saudi Arabia about Families & Homes. The parameters are grouped into three macro-parameters: Nurturing Families, Resources & Management, and Challenges. We provide an overview of the parameters and taxonomy in Section 5.1. The quantitative analysis is discussed in Section 5.2. Subsequently, we discuss each macro-parameter in separate sections, Sections 5.3–5.5

### 5.1. Overview and Taxonomy

Our Bert Topic modeling detected a total of 226 clusters and reducing it to 25 clusters based on the Arabic twitter dataset. Then after the analysis of all the clusters we found 3 clusters are irrelevant and we exclude them from our analysis for different reasons. The cluster 14 was removed since it was about hate speech against particular people, while clusters 15 and 18 were excluded due to their unclear themes. The remining is 22 clusters called parameters. The parameters were grouped into three macro-parameters based on our domain knowledge, similarity matrix, hierarchical clustering, and other quantitative methods. In Section 3, we discussed the methodology and process used to discover parameters and group them into macro-parameters.

Table 7 lists the parameters and the macro-parameters detected by our BERT model from the Arabic twitter dataset with its top 10 keywords. The keywords are sorted according to their importance score. We grouped the parameters into three macro-parameters represented in column 1. These are Nurturing Families, Resources & Management, and Challenges. The parameters and its IDs listed in column 2, 3 respectively. The fifth column highlights the top 10 keywords associated with each parameter. This list of keywords is primarily reflected and represent the theme of the parameter. As part of our effort to gain a better understanding of the parameters, we examined the tweets associated with each parameter. As shown in the following table, the Arabic keywords are listed along with their English translations. We have also contextually translated the Arabic tweet's content so that English readers can better understand the content.

The parameters detected by our tool were used to build a taxonomy (see Figure 18) describing Families & homes. The taxonomy shows the parameters, their macro-parameters. The first level represents the macro-parameters Nurturing Family, Resourcing & Managements, and Challenges. Every macro-parameter contains many parameters. The second level branches represent these parameters e.g., Gratitude, Education & Society, Nurturing Family Values, and Seeking Marriage, etc.

**Table 7.** Macro-Parameters and Parameter for Families & Homes (Data Source: Twitter).

| Macro | Parameter | ID | Keywords |
|---|---|---|---|
| رعاية الأسرة Nurturing Families | إمتنان Gratitude | 0 | نعمه، الناس، بيوتنا، الوطن، ابناء، الحياه، نقدر، صغار، الدنيا Blessing, People, Our Homes, Homeland, That We, Sons, Life, Appreciate, Young, World |
| | التعليم والمجتمع Education & Society | 8 | التربيه، وزاره، العربيه، الأطفال، الاجتماعيه، المملكه، المدرسه، الابتدائيه، التعليميه، المجتمع Education, Ministry, Arabic, Children, Social, The Kingdom, School, Primary, Educational, Society |
| | البحث عن الزواج Seeking Marriage | 23 | سعوديه، جميله، الجنسيه، موظفه، البشره، الوظيفه، لايوجد، بزواج، الزوجيه، الاجتماعيه Saudi, Beautiful, Nationality, Employee, Complexion, Job, Not Exist, With Marriage, Marital, Social |
| | رعاية قيم الأسرة Nurturing Family Values | 13 | رجعت، رجعوني، رمضان، رعايه، رعايه، تتملك، روحي، وحفظه، الوالدين، راجعه، رياض Get Back, Bring Me Back, Ramadan, Take Care, Own, Take Possession of My Soul, Preserve It, Parents, Take It Back, Riyadh |
| | الترابط الأسري Family Cohesion | 6 | بيتنا، البيت، اخوات، العائله، عائلتي، بيتك، العايله، مسكن، الأطفال، أبناء Our House, House, Sisters, Family, My Family, Your House, Your Family, Dwelling, Children, Sons |
| | رعاية الأطفال Nurturing Children | 2 | شخص، طفل، تقول، بيتك، طيب، شغل، البنت، تعرف، دخل، ممكن Person, Kid, Say, Your Home, Ok, Work, Girl, Know, Enter, Possible |
| | الرفقة الحسنة Good Companionship | 20 | بيتنا، سطح، سوريا، صاير، صرت، صديق، صاحب، صوره، اطفال، صغار Our House, Roof, Syria, Became, I Became, Friend, Owner, Picture, Children, Young |
| | الزيارات العائلية Family Gatherings | 17 | دخل، بيتنا، دائما، اخوات، داخله، دكتور، دكه، دقايق، مجلس، العيله Entered, Our House, Sisters, Doctor, Bench, Minutes, Sitting Room, Family |
| | الاجتماع بالجدة Gatherings with Grandmothers | 9 | جدتي، جديد، جنه، جعلنا، جمعه، جاك، جاب، جميله، جمالك، جلسه Grandmother, New, Paradise, We Made, Gather, Come to You, Bring, Beautiful, Your Beauty, Gathering |
| | أنشطة ترفيهيه Entertainment Activities | 10 | فعاليات، فخر، فديت، بيتكم، فوازير، فاتنه، فطار، فقدان، فلتحيا، بالعز Events, Pride, Redeemed, Your House, Riddle, Beautiful, Breakfast, Loss, Long Live, With Honor |
| الموارد والإدارة Resources & Management | تمويل المنزل House Financing & Affordability | 7 | تتملك، تمويل، تملك، تربيه، تمويلك، تمول، طلعت، عقار، تعالي، تويتر Own, Finance, Hold, Raise, Investment, Your fund, Went Out, Real Estate, Come, Twitter |
| | | 11 | تنظيف، خزانات، سجاد، منازل، مفروشه، وحديثه، مكيفات، التشطيب، مسابح، استراحات Cleaning, Tanks, Carpets, Homes, Furnished, Modern, Air Conditioning, Finishing, Swimming Pools, Rest Houses |
| | تنظيف المنزل Home Cleaning | 16 | التنظيف، الموجوده، خدماتنا، شبابيك، موكيت، الواجهات، بافضل، حسابنا، كنب، منازل Cleaning, Existing, Our Services, Windows, Carpets, Interfaces, Best Way, Our Account, Sofa, Homes |
| | | 19 | غريبه، بساطك، غرفتي، ترتيب، مزعج، غبار، غابت، غازي، غبنه، غداي Strange, Your Rug, My Room, Organize, Annoy, Dust, Absent, Gazi, Overcome, Lunch |
| | | 21 | مكيفات، منازل، خزانات، مبيدات، شركه، مكافحه، بالخرج، نظافه، للتنظيف، فراشات Air Conditioners, Homes, Tanks, Pesticides, Company, Combat, In Al-Kharj, Cleaning, For Cleaning, Butterflies |
| | مكافحة الحشرات Pest Control | 22 | مكافحه، الحشرات، مبيدات، الفئران، تركيب، الذهبيه، المبيدات، صراصير اشواك، العقارب Anti, Insects, Pesticides, Mice, Installation, Golden, Pesticides, cockroaches, Thorns, Scorpions |
| | تنظيف المسابح Swimming Pool Cleaning | 24 | تنظيف، منازل، خزانات، الذهبيه، تعقيم، شركه تنظيف، السيراميك، بالمدينه، مسابح، والحناكيه Cleaning, Houses, Tanks, Golden, Furnished, Cleaning Company, Ceramics, In the City, Swimming Pools, The Hanakia |
| تحديات Challenges | العمل والتراخي Work & Indolence | 1 | شغل، شخص، شكلي، شوارع، شيخ، شجره، شركه، شوفي، تنظيف، بيتك Work, Person, Formal, Old, Streets, Tree, Company, Shop, Cleaning, Your Home |
| | الخمول البدني Physical Inactivity | 5 | البيت، بيتنا، بيتي، طفل، برا، سنة، الأطفال، الف، ساعه، الوداد Home, Our Home, My Home, A Child, Outside, A Year, Children, A Thousand, An Hour, Al Wedad |
| | عادات النوم Sleeping Habits | 3 | نعمه، الكهف، نرجع، صغار، نروح، بيوتنا، نورت، نحنا، نومه، نشوف Grace, The Cave, We Go Back, Young, We Go, Our Homes, We Light Up, We Sleep, We See |
| | التحديات الاجتماعية والاقتصادية للمرأة Socioeconomic Challenges for Women | 12 | زواج، مطلقه، سعوديه، جميله، موظفه، البشره، الوظيفه، قبيليه، الجنسيه، الاجتماعيه Marriage, Divorced, Saudi, Beautiful, Female Employee, Complexion, Job, Tribal, Nationality, Social |
| | كورونا COVID-19 | 4 | الأطفال، البيوت، بيوتنا، الآخره، أنزله، الصديقين، مراتب، الأرزاق، والشهداء، كورونا Children, Homes, Our Homes, The Hereafter, Sent Down, The Two Friends ,Ranks, Livelihoods, Martyrs, Corona |

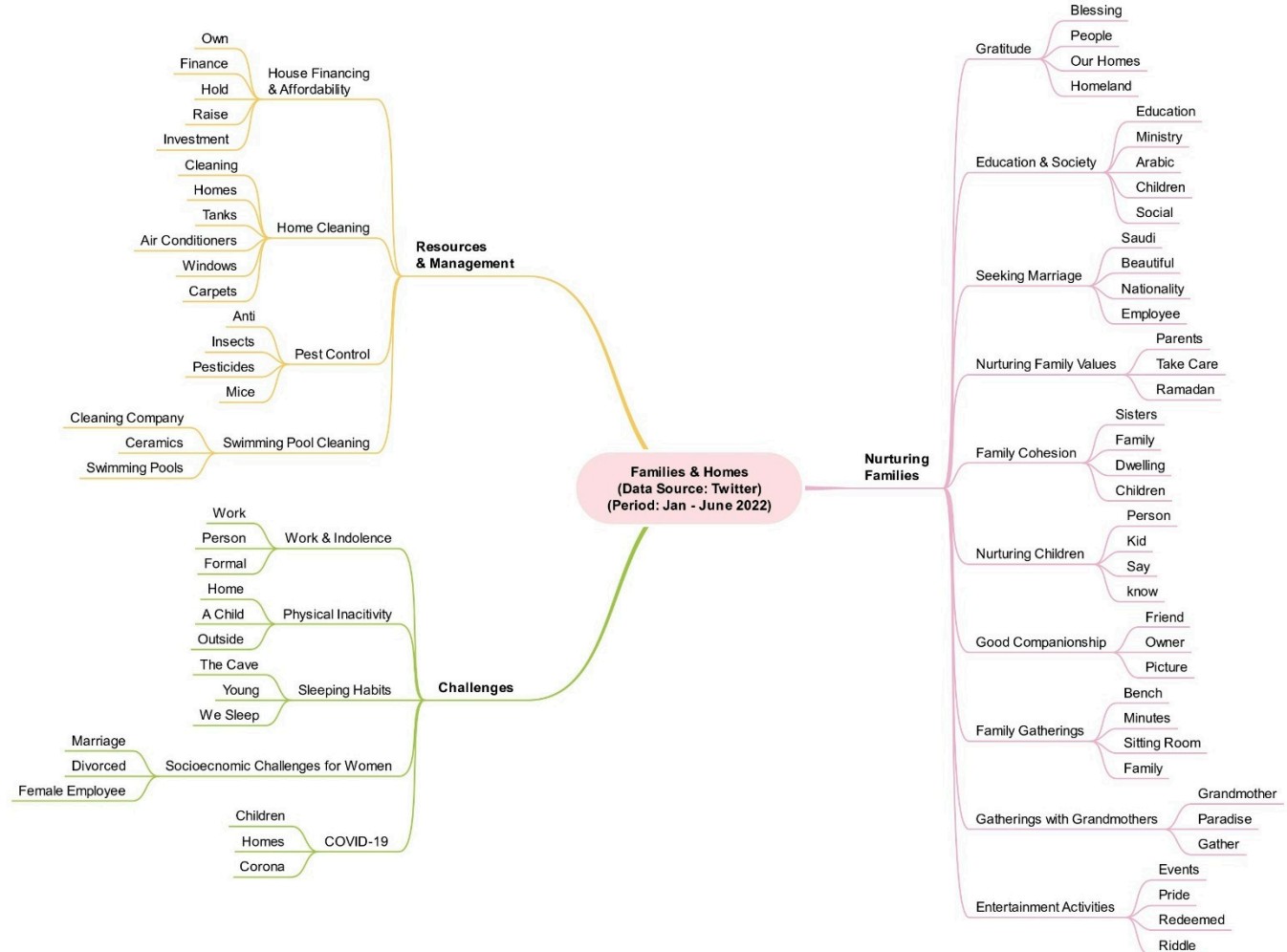

**Figure 18.** Taxonomy of discovered Home and Families parameters extracted from Twitter data.

*5.2. Quantitative Analysis*

This section outlines term scoring, word scoring, inter-topic distance maps, hierarchical clustering, and similarity matrices. Almost all parameters are represented by a group of ten keywords that accurately describe that parameter.

Figure 19 visualize the top 10 keywords for each parameter (see Section 3.9). The importance score, or c-TF-IDF, is used to order the keywords. There are 25 subfigures, and, in each subfigure, the horizontal line shows the importance score, and the vertical line shows the parameter keywords. The bars are in different colors to help differentiate between them; the colors do not have any specific meaning or representation.

As shown in Figure 20, the inter-topic distance map is based on a multidimensional scale, in which six groups of parameters are identified (see Section 3.9). D1 and D2 represent the two dimensions. The bottom left group has a larger parameter size than the other groups. On the right side, there are three parameters of small size. Additionally, there are two clusters of approximately the same size around the middle of the map. However, the parameters were manually grouped into three macro-parameters.

Figure 21 describes the hierarchical clustering of the 25 parameters and systematically pairs them based on the cosine similarity matrix (see Section 3.9).

Figure 22 visualizes the similarity matrix among the parameters (see Section 3.9). Light green color represents the least similarity between parameters, while dark blue represents the highest similarity.

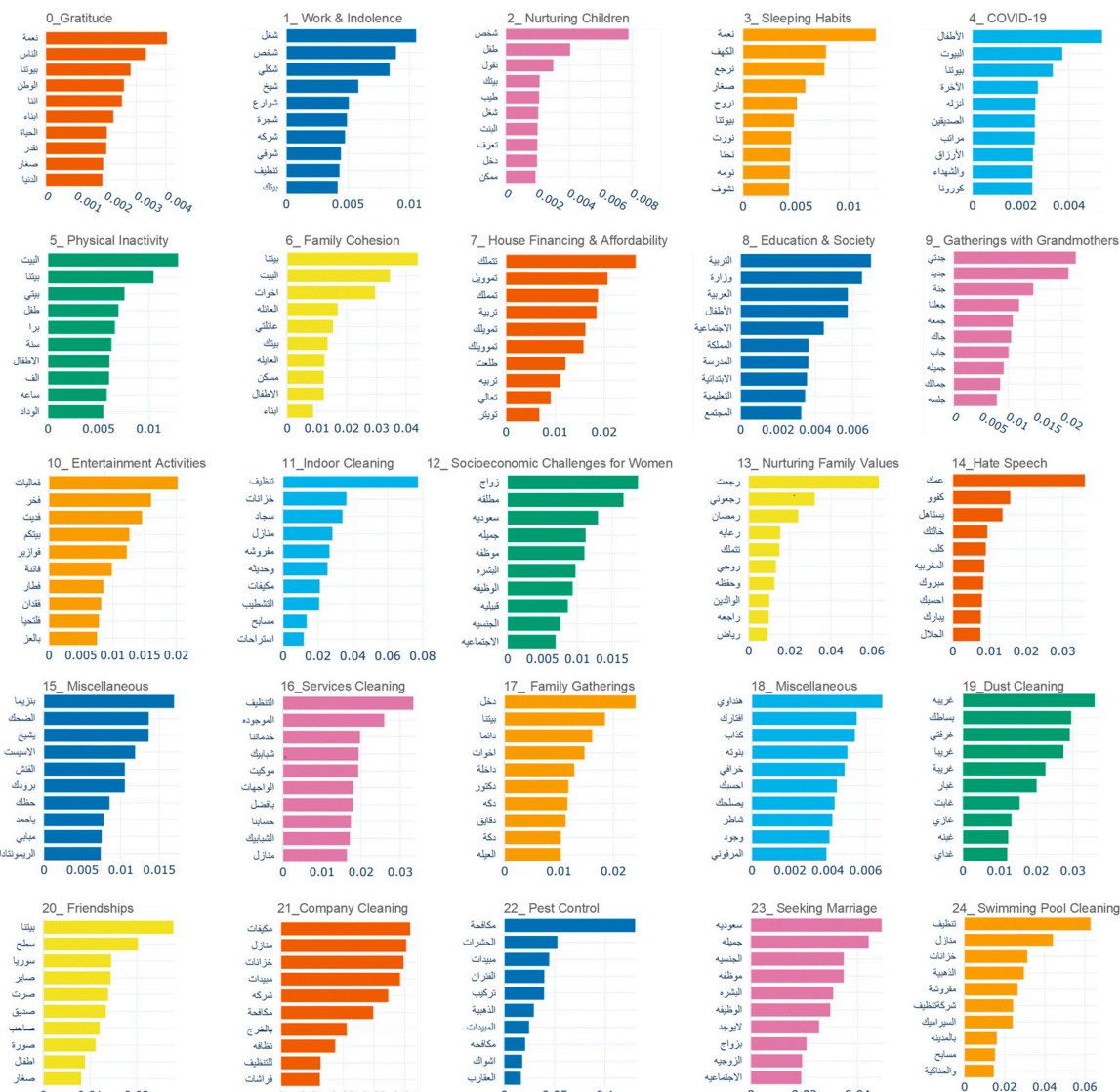

**Figure 19.** Twitter parameters with keywords c-TF-IDF score (Data Source: Twitter).

*5.3. Nurturing Families (Twitter)*

We begin discussing the parameters related to the first macro-parameter, Nurturing Families. It includes ten parameters: Gratitude, Education & Society, Seeking Marriage, Nurturing Family Values, Family Cohesion, Nurturing Children, Good Companionship, Family Gatherings, Gatherings with Grandmothers, and Entertainment Activities. The first parameter is Gratitude, represented by key terms such as Blessing, People, Our Homes, Homeland, That We, Sons, Life, Appreciate, Young, World. The tweets related to this topic, for example

"هذا التاريخ العميق يستحق أن يستذكره أبناء هذا الوطن."

*"This deep history deserves to be remembered by the people of this country."*

The second parameter is Education & Society. This parameter was referred to by keywords such as Education, Ministry, Arabic, Children, Social, The Kingdom, School, Primary, Educational, Society. There are tweets related to this parameter, for example

"...الأُمُّ الصالِحةُ إذا صلَحَتْ صلَحَ حال الأُسْرة؛ فصلَحَ المجتمعُ بأسرِه."

*"Good mother is good for the family; when she is good, the whole society is reconciled."*

"... التربية و التعليم عبارة عن مجموعة من التحولات الايجابية المركبة والمتنوعة التي يلحظها الاستاذ بدقة ولا ينتبه لها اولياء الامور الا من رحم ربي."

" . . . *Education is a series of complex and varied transformations that the teacher observes precisely but that parents might ignore.*"

The third parameter is Seeking Marriage. Several keywords are used to describe it, including Saudi, Beautiful, Nationality, Employee, Complexion, Job, Not Exist, With Marriage, Marital, Social. In our dataset, this parameter represented by the following tweets.

"الزواج مو هدف لكن تحقيقه وتكوين اسره شي جميل ويسبب سعاده."

"*Marriage is not a goal but achieving it and starting a family is beautiful and causes happiness.*"

"الزواج تعاون و مشاركة، في حال اذا كانت الزوجة تعمل، الزوج يقوم بواجبات المنزل و العكس صحيح، في هذه الحالات العناد يسبب الإهمال."

"*In a marriage, there is cooperation and partnership, if the wife works, the husband performs the duties of the house, and vice versa, in these situations, stubbornness can cause neglect.*"

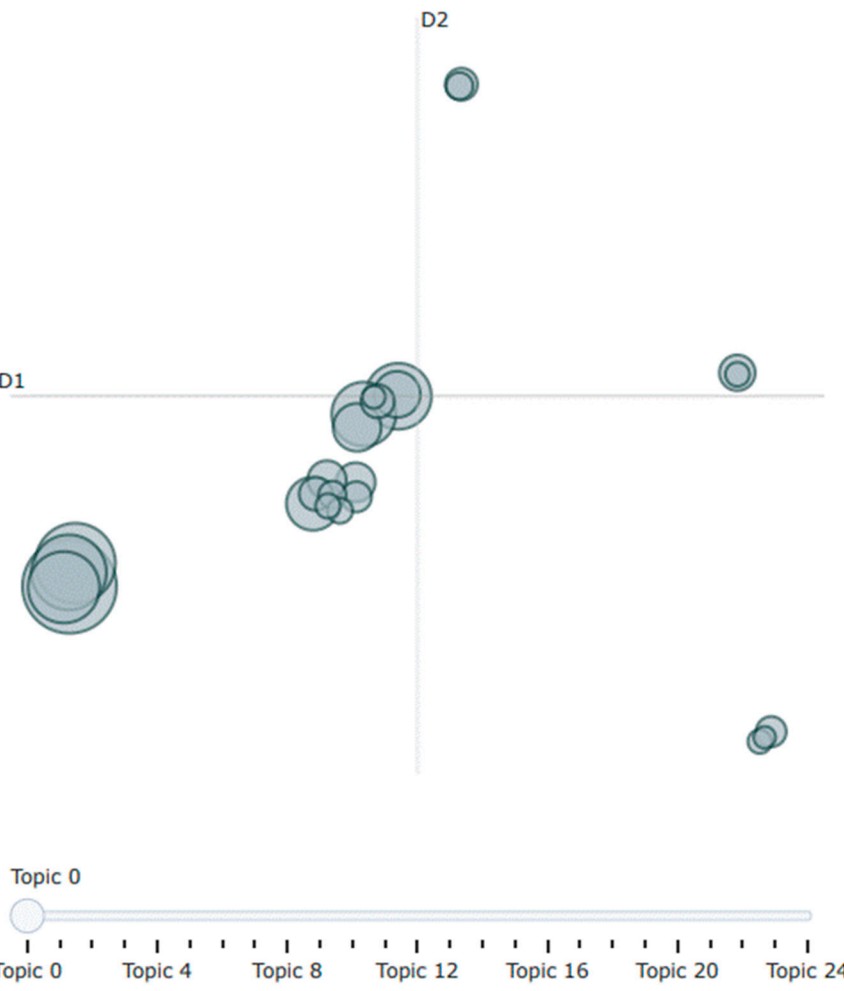

**Figure 20.** Inter-topic distance map (Data Source: Twitter).

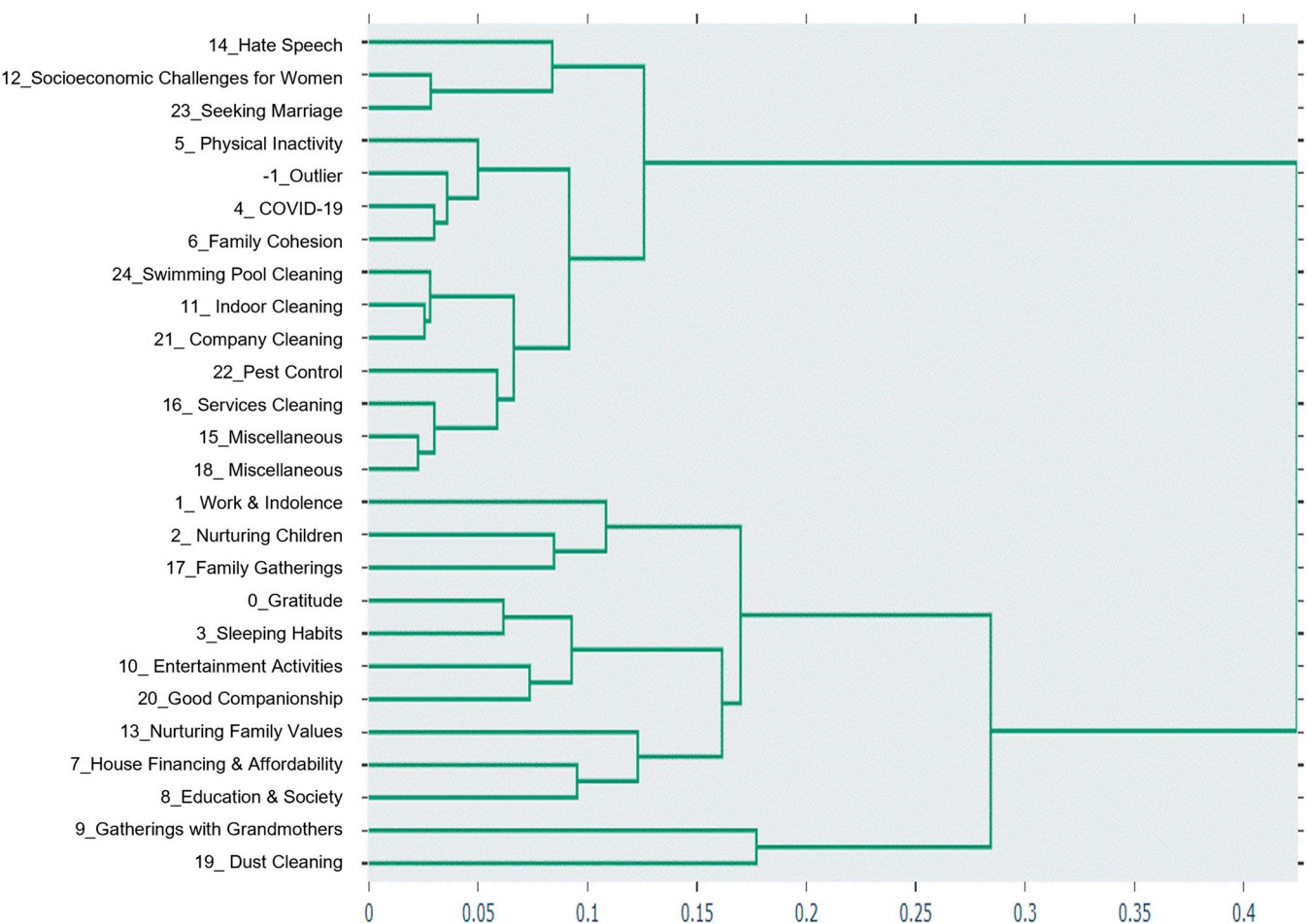

**Figure 21.** Hierarchical clustering (Data Source: Twitter).

Families are the building blocks of a better society and a stronger nation. People believe that once the family is set right, society will be able to find its way toward unity and progress. Nurturing Family Values is the main thing to building a better and strong family. Our fourth parameter is Nurturing Family Values, represented by keywords such as get back, bring me back, Ramadan, take care, own, take possession of my soul, preserve it, parents, take it back, Riyadh. A sampling of related tweets is provided below

"كلما زادت الصدقة زاد الرزق، كلما زاد الخشوع في الصلاة زادت السعادة، كلما. زاد بر الوالدين زاد التوفيق بحياتك"

*"Charity brings sustenance, reverence in prayer correlate with happiness, if you honor your parents, your life will be more successful."*

"البنت لا تعرف مسؤولية البيت وتحمل رعاية أسرة... علموا بناتكم ان الأسرة بناء يحتاج إلى أسس لابد أن نتعلمها. فالاسرة مسؤولية كبيرة."

*"The child does not realize the responsibility of the house and the care of a family . . . It is important to teach your children that the family is a building that must have foundations. Family is a great responsibility."*

"فيه ناس طموحها تصنع عائلة، وناس تشوف السفر والترحال نمط حياة، والناس تشوف التفوق بالشهادات،وناس حتى قومتهم من السرير ونفض الكسل. يشوفونه انجاز"

*"The ambition of some is to create a family, another is to travel, and others see excellence in diplomas. People even see getting up from bed and overcoming laziness as an achievement."*

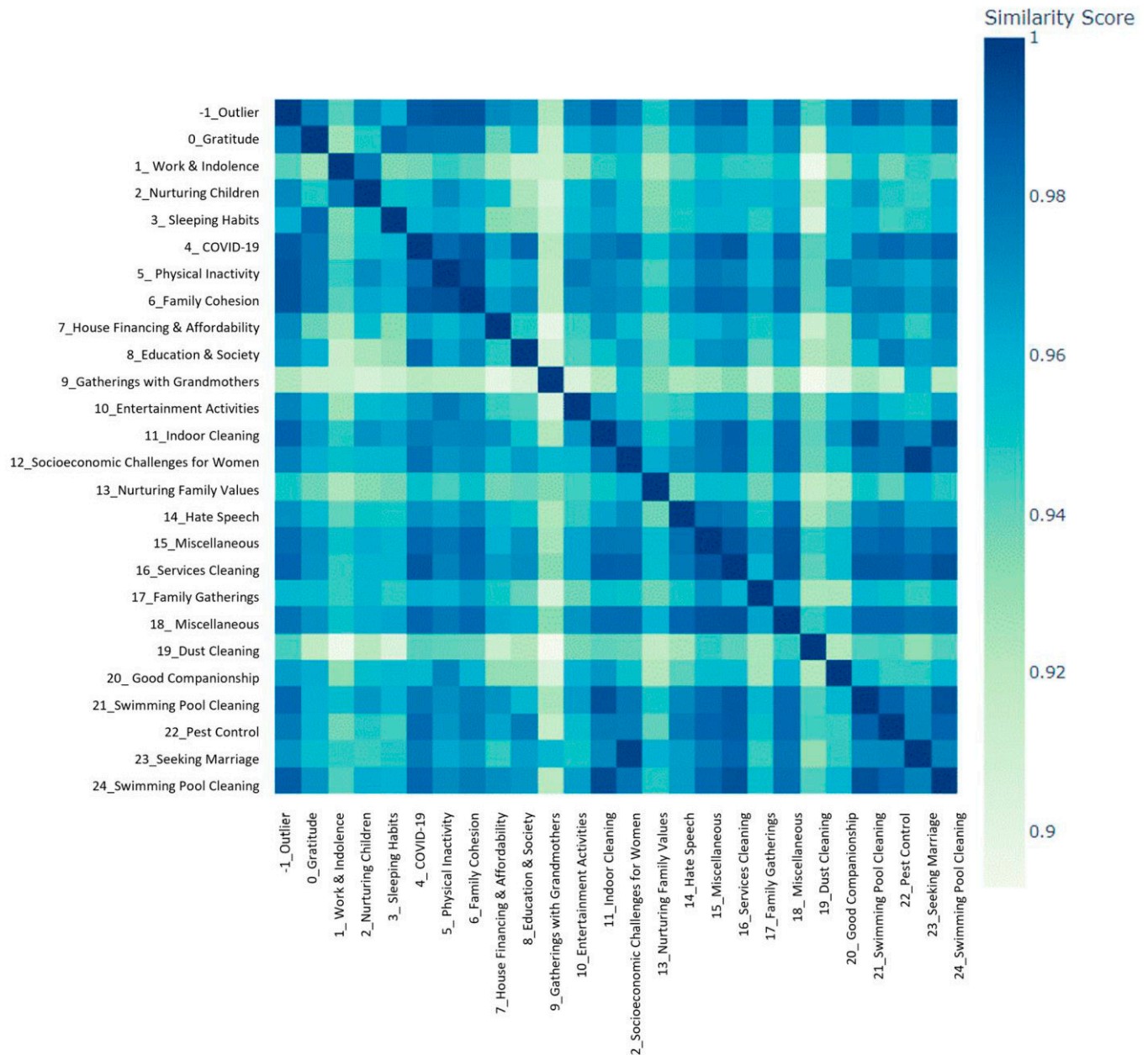

**Figure 22.** Similarity matrix (Data Source: Twitter).

Family Cohesion is the fifth parameter. The list of keywords that describe it are Our House, House, Sisters, Family, My Family, Your House, Your Family, Dwelling, Children, Sons. A lot of tweets, found in our dataset, related to this parameter were like the following tweets

"اللهم عائلتي ومن أحب حتى ظلال الجنه".

*"Please God, my family and those who I love in Heaven"*

"صوت الوالدين فالبيت أجمل وأكبر نعمة".

*"Parents' voice, home is the most beautiful and greatest blessing."*

"الأخت الكبيرة هي جمال كل بيت، تُشعرك بالسعادة اللهم احفظها".

*"The big sister is the beauty of every home, she makes you feel happy, may God protect her."*

"مهما كانت درجة قربك من اي حد سواء اخت. اخ. صاحبة او اي احد ثاني هتفضل الأم هي الأقرب وأكثر شخص ممكن يحس بيك ويفهمك وصدقني اللي ممكن تحكيه لوالدتك ما ينفعش تحكيه لاحد ثاني مهما كان."

*"It doesn't matter how close you are to your sister or brother. The mother is the closest and most understanding person, and believe me, what you can tell your mother, you won't be able to tell anyone else, no matter what, no matter what."*

The sixth parameter is Nurturing Children. It is characterized by keywords such as person, kid, say, your home, ok, work, girl, know, enter, and possible. The following are tweets related to this parameter that discuss raising children under a set of values, developing personality traits in children, etc.:

"في ظل انتشار وسائل التواصل ومعها المشاهير اصبح ضرورة تربية الأبناء على الاعتزاز بالقيم وتقوية شخصياتهم".

*"In light of the proliferation of social media and celebrity culture, it has become increasingly essential to raise children to be proud of the values and strengthen their personalities"*

"الشخص يتفاعل مع البيئه اللي نشأ فيها و يطور مهاراته من خلال المشاكل اللي تواجهه خصوصا لو كان طفل. الامهات دايم حريصات بزياده اكثر من الاباء و الشخص الحريص على ولده بزياده ولا يخليه يحل مشاكله يطلع ولده مايفهم ماهو ذنب الطفل لكن ذنب الاب اللي حرمه حقه من التعلم من مشاكل الحياه."

*"A person interacts with his environment and develops his skills based on the challenges he faces, especially as a child. Mothers always put more effort into that than fathers, and if a father does not allow his son to solve his problems, his son will not develop a strong personality. It may not be the child's fault, but the father's who may have deprived him from his right to learn from life's problems."*

"في مرحلة بناء شخصية الطفل، اذا تدخلتي في هذي المرحلة بكره راح يغلط على اي شخص و يتصفق و يجي يكلمك. لازم يعرف كيف انه يتجنب الناس هذي وكيف يتعامل معهم ...."

*"As the child is building his personality, if you interfere in this stage, tomorrow he will make a mistake, be hit, and come to talk to you. He must learn by practice how to avoid and deal with these people . . . "*

"البنت اللي اهلها معطينها الحب والثقه دايماً بتكون شخصيتها جداً قوية ، فعلاً ان وراء كل بنت عظيمة أهل عظيمين جد"

*"A girl whose parents give her love and trust will always have a very strong personality. There are very great family behind every great girl."*

Good Companionship is the seventh parameter for the macro-parameter Nurturing Families. It described by a set of keywords as our house, roof, Syria, became, I became, friend, owner, picture, children, and young. A few examples of tweets are listed here

..."قد تُكتسب الأخلاق الحسنة بمصاحبة أهل الخير، فإن الطبع لصٌّ يسرق الخير والشر"...

*"Good manners may be acquired by accompanying good people, . . . "*

"انصحك اذا تبغين حياه سعيده ودائمه واستقرار بحياتك. اقطعي علاقتك تماما بهالصديقات وغير مأسوف عليهم"

*"I advise you if you want a happy and lasting life and stability in your life. Completely cut off your relationship with bad friends and do not regret them."*

The eighth parameter is Family Gatherings, represented by the keywords Entered, Our House, Sisters, Doctor, Bench, Minutes, Sitting Room, Family. Following are tweets posted related to this parameter.

"منظر ضحكة العائلة بشكلٍ جماعي يستحق التخليد في إطار. يعلق في زوايا القلب."

*"The sight of the family laughing together deserves to be immortalized in a frame. It will always remain in my heart."*

"سعادة الأسرة والأقارب والأهل وحلاوة الزيارات العائلية اجمل بكثير من لوثة العلاقات المبنية على مصلحة. "

*"The sweetness of family visits and the happiness of family, relatives, and parents is much more beautiful than the pollution of interest-based relationships."*

The ninth parameter is Gatherings with Grandmothers. It includes keywords such as grandmother, new, paradise, we made, gather, come to you, bring, beautiful, your beauty, gathering. Here are examples of tweets

"الجدّة نُور كل بيت و روح العائلة ، ياربّ احفظ جدتي."

*"Grandmothers are the light of every home and the soul of every family. Lord, protect my grandmother."*

"وتبقى الجده هي الحُب الذي لاينتهي ،عكاز كل بيت ونور كل بيت وروح العائلة ،يارب جدتي في ودائعك وحمايتك . "

*"Grandmother remains the pillar of every home, the light of every home, and the soul of every family"*

The last parameter is Entertainment Activities. This parameter is referenced in the following tweets

"في آرت بروميناد الكل سعيد ومستمتع بفعالياته اللي تناسب العائلة والأصحاب ."

*"At . . . , everyone is happy and enjoying its activities that are suitable for family and friends."*

"صار وقتنا في البيت مع العائلة أكثر وعرفنا دفء البيت وقيمة عوائلنا.. ويومنا خفيف وكله فعاليات و اكل و مسلسلات وضحك وسوالف و اكتشفنا اهتماماتنا وصار عندنا هوايات وحاجات جديدة في حين كان الدوام أكبر عائق."

*"As we spent more time at home with our families, we became aware of the warmth of the home and the value of our families. We have a light day with all events, eating, series, laughter, and sideburns, and we discovered our interests and needs, while work was the biggest obstacle."*

" ضمن فعاليات ملتقى الأم والطفل فلذات تم بحمد الله الانتهاء من دورة فن التعامل مع الأطفال"...

*"Within the activities of the Mother and Child Forum, the course of the art of dealing with children has been completed."*

*5.4. Resources & Management (Twitter)*

Next, we discuss the second macro-parameter Resources & Management. It involves the tweets and parameters that are related to home finance and clean. It includes four parameters; the first parameter is House Financing & Affordability. It includes keywords own, finance, hold, raise, investment, and fund. An example of a tweet is:

" اقتصد من راتبك الذي تستطيع وادخل في جمعيات كي تمسك المبلغ سنه على سنه تستطيع ان تجمع مبلغ افضل لك من البنوك "وبعيد عن الاقتصاد وبعدها خذ التمويل العقاري مع المبلغ الذي جمعته ...

*"Save as much money as you can from your salary and join associations to keep it year after year. You can collect a better amount from those banks give and away from the economy, and then take the real estate financing with the amount you collected . . . "*

The second parameter is about Home Cleaning. It was created by merging four parameters 11, 16, 19, and 21. All these parameters discussing general home cleaning for tanks, carpets, homes, furnished, air conditioners, windows, and rooms. The third parameter is Pest Control, and the fourth is Swimming Pool Cleaning. Examples of tweets related to these parameters are listing in the following

"شركة تنظيف شقق فلل بالرياض تقدم خدمات بمنتهي الدقة في تنظيف شقق مجالس فلل و غسيل خزانات فرشات زوليات"

*"A company cleaning villas and apartments in Riyadh provides very meticulous services in cleaning villa council apartments and washing tanks, carpets, and mattresses"*

"تعقيم مكافحة حشرات رش مبيدات نظافة عامة"

*"Insect control sterilization, spraying general hygiene pesticides"*

"تنظيف مجالس كنب موكيت سجاد ستائر تنظيف خزانات مسابح"

*"Cleaning of sofas, carpets, curtains, cleaning swimming pool tanks."*

*5.5. Challenges (Twitter)*

Our last macro-parameter, Challenges, refers to what family members experience in terms of challenges. It consists of five parameters. Work & Indolence, Physical Inactivity, Sleeping Habits, Socioeconomic Challenges for Women, and COVID-19.

Work & Indolence parameter includes the list of keywords Work, Person, Formal, Old, Streets, Tree, Company, Shop, Cleaning, Your Home. Examples of tweets are

"عاادي الرجال مايعيبه الا الجلسه في البيت بدون عمل اما الشخص اللي يشتغل ويكد على بيته واطفاله ومكفيهم الحاجه لايضره بل فخر له مهما كانت الوظيفه."

*"The only thing wrong with men is sitting at home without working. For a person who works hard for his family to provide for their needs, it does not harm him, rather it makes him proud, whatever his job may be."*

"الحمدلله على نعمه الشغل من البيت بس اشتقت للمكتب."

*"It is a blessing to work from home, but I miss the office."*

Physical Inactivity is about people complaining about physical inactivity of children mainly and adult because they are addicted to playing game and watching football, etc. This parameter is represented by the keywords home, our home, my home, a child, outside, a year, children, a thousand, an hour. The following tweets were posted:

"كيف أقلل من وقت جلوس أطفالي على الشاشات؟"

*"How do I reduce my children's screen time?"*

"الطفل الطبيعي يجب أن يحب الخروج من المنزل، ويحب الاجتماعات... بعض الأمهات تفخر بأن طفلها يحب الجلوس في البيت"

*"A normal child should like to go out of the house, and like meetings . . . Some mothers are proud that their child likes to sit at home."*

The Sleeping Habits is the third parameter for the Challenges macro parameter. There are some keywords that pertain to it: grace, the cave, we go back, young, we go, our homes, we light up, we sleep, we see. Several tweets about this parameter are listed below.

"نومي نوم اهل الكهف"

*"My sleep is the sleep of the people of the cave."*

"نوم الليل ودعته مع الأمومة"

*"The night's sleep is over because of motherhood"*

"شاهد أفضل وضعية للنوم لمن يعانون من آلام الظهر"

*"See the best sleeping position for back pain sufferers"*

The fourth parameter is Socioeconomic Challenges for Women. As part of the Saudi Vision of 2030 and the National Transformation Program 2020, women with various social statuses in the Saudi Arabian Kingdom are supported. This parameter is represented by keywords such as marriage, divorce, Saudi, female employee, tribal, nationality, socially and others are used to represent this. Listed below are some examples of related tweets

"انتبهو تفرطون في : (الشهادة الجامعية، الوظيفة، الاستقلال المادي) نصيحتي لبنات العايلة."

*"My advice to the daughters of the family is to not lose university degree, job, financial independence"*

"المرأه العاقله المتزنه لايهما من يحاول ان يهدم كيانها هي تعلم بانها سيدة البيت ولها كل احترام وتقدير وحريه فيما يكون لها مناسب ويرفع قيمتها و مكانتها الاجتماعية."

*"Women who are sane and balanced are aware that they are the masters of their houses and have all the respect, appreciation, and freedom that increases their value and social status."*

The last parameter for this last macro-parameter is COVID-19. It is about supporting people in taking the vaccines and taking all precautions against the Coronavirus. Furthermore, it highlights COVID-19 deaths and wishes well to those who lost their lives to it. The following are a few tweets relating to this parameter.

"...: كورونا يصيب الأطفال وينتشر بينهم وتطعيمهم باللقاح ضرورة"

*" . . . Corona infects children and spreads among them, and vaccinating them with the vaccine is a necessity"*

"أثبتت وزارة التعليم قدرتها على مواجهة التحديات في ظل جائحة فيروس كورونا ، واستمرار العملية التعليمية بجهود وإخلاص من كافة العاملين في الميدان التعليمي وبشراكة مع الأسرة والمجتمع"

*"The Ministry of Education has proven its ability to face challenges in light of the Corona virus pandemic, and the continuation of the educational process with the efforts and sincerity of all workers in the educational field and in partnership with the family and society."*

## 6. Discussion

In our work, we aim to identify families and homes parameters through a data-driven parameter discovery methodology of two different perspectives: the academic view and the public view from two different data sources, English Scopus and Arabic Twitter. Details of these two types of families & homes parameters can be found in Sections 4 and 5. Our developed methodology can be applied to a wide range of subjects and topics.

We discovered a total of fifty families and homes parameters from the Scopus academic dataset and grouped them into five macro-parameters: Nurturing Families, Health & Lifestyle, Communities & Nations, Resources & Management, and Technologies.

The first macro-parameter Nurturing Families, touches upon establishing families & homes [70,76,88,94,97]. It captures research related to the children's mental, physical, social, and academic development, including their parent's involvement. The roles, responsibilities, challenges, and issues concerning women while developing, maintaining, and taking care of their families and homes. In addition to the elderly people's related topics involving their lifestyle, health, psychological conditions at care homes, and various aspects of the family spirituality including religion, faith, religious education, and religion-related social studies. The Health & Lifestyle macro-parameter captures parameters related to lifestyles, diseases and healthcare provision [104,106,118,121,145], for example, the effects of genetic characteristics, food consumption and eating habits, obesity prevalence, causes, treatments, and prevention, smoking's effects, addictions, and drug abuse factors, supporting and dealing with mental conditions, cancer patients, detection and social support, diabetes-related research, healthcare provisioning and developing healthcare professionals, homecare and nursing homes, and Pandemics effects. The third macro-parameter Communities & Nations

covers various dimensions of families and homes distinguished by different communities, countries, societies, or nations [154,160,196,204,216]. It captures parameters related to different racial groups, LGBTQ communities, various political affiliations, Mexican immigrants, Canadian people and immigrants in Canada, Syrian Immigrants, and other countries such as Germany, Sweden, Finland, Italy, Spain, China, Japan, and Australia.

The macro-parameter Resources & Management concerns about establishing home resources and managing families and homes [219,225,231,240,251]. It discusses aspects related to families and homes economics and operations. Examples of academic research captured by this macro-parameter include research regarding estate prices and different factors that affect pricing and family affordability. Studies related to house design include architectural design, archaeology of houses, antiquities, sites and locations, materials and techniques, and some specific heritage homes. Research related to jobs, work environment, work-family conflicts, family business, and families involved in farming and agriculture. Moreover, it captures dimensions related to tourism and its effects, opportunities, issues, and challenges with families and homes. This macro-parameter has also captured energy and water resource management and the effects of floods, earthquakes, and other disasters on families and homes. The fifth macro-parameter Technologies captures parameters related to the utilization of technology in various aspects related to the families, homes, and individuals to improve the quality of life [271,276,282,286,289]. Parameters under this macro-parameter discuss assistive robots, remote healthcare, social media, smart environments and IoTs, sleep monitoring, and home-based training and rehabilitation systems. A detailed discussion of these topics is provided in the Parameter discovery for Families & Homes (Academia: Scopus) section, along with references that support it.

There are a wide variety of topics covered by academic research on families & homes, such as nursing & homecare, women, children development, elderly, smart environments technologies, assistive robots, family business, and so on. Figure 23 shows the word cloud of the academic parameters' keywords determined by BERT modelling, where the size of the keywords represents their frequency, which is linked to their importance. The keywords like home, family, social, parent, health, care, and child are highly related to the families & homes field.

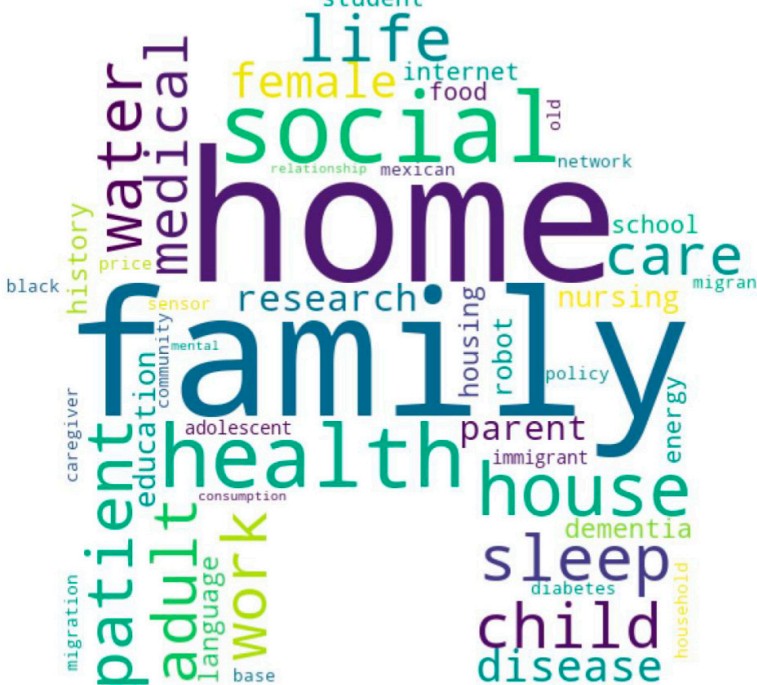

**Figure 23.** A Word Cloud Generated from the Families & Homes Parameter Keywords (Data Source: Scopus).

From the Twitter dataset, 25 parameters were identified and grouped into three macro-parameters: Nurturing Families, Resources & Management, and Challenges. The word cloud shoes the families & homes parameters keywords discovered by our BERT modelling from the Twitter dataset. There is a detailed discussion of these macro-parameters in Section 5, along with tweets supporting the discussion. Figure 24 depicts the word cloud of the parameters' keywords where each keyword's size indicates how frequently it is found. Public families & homes discussion covers a wide range of topics, such as nurturing children, family values, home cleaning, family relationship, employment, etc.

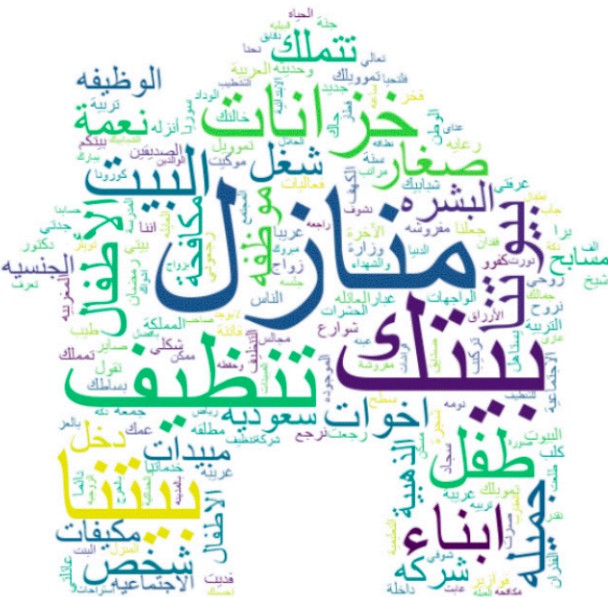

**Figure 24.** A Word Cloud Generated from the Families & Homes Parameter Keywords (Data Source: Twitter).

Based on our data-driven families and homes parameters discovery, we introduce a data-driven parameter discovery, decision making and implementation approach aiming to improve families. A high-level framework showing our proposed data-driven approach for smart families and homes is provided in Figure 25. It shows that various actors work on identified challenges to achieve and reach families and homes objectives by using methods and solutions. Our framework can identify challenges, methods and solutions, and objectives that aim to benefit and improve families. The high-level or ultimate objectives are to enable better smart families and homes. Example of objectives includes happiness, healthy people, preventing violence, and others listed in the figure. The stakeholders are children, adolescents, parents, and others. The challenges may include education and affordability, parenthood and career balance, etc., technologies, awareness of cultures and communities, sustainable practices, strengthening families, healthy lifestyles and good habits are considered enablers, methods, and solutions.

This work makes significant theoretical and practical contributions and support and extends the earlier works that stressed the need for defining homes [5,6]. This work also extends the Deep Journalism approach [38] that we used to discover multi-perspective parameters for transportation using three data sources, The Guardian, Web of Science, and Traffic Technology International Magazine. The three data sources provided the public, government, industry, and academic perspectives on transportation. This work and Deep Journalism [38] are continuations and improvements to our earlier works in this research direction on parameter discovery from Twitter data for education and learning during the COVID-19 pandemic [39] and general governance measures during COVID-19 [46], the discovery of healthcare services for cancer [40], and detection of transportation events [43,44] and diseases [2]. This work, therefore, also provides evidence to support the general literature on using Twitter as a data source.

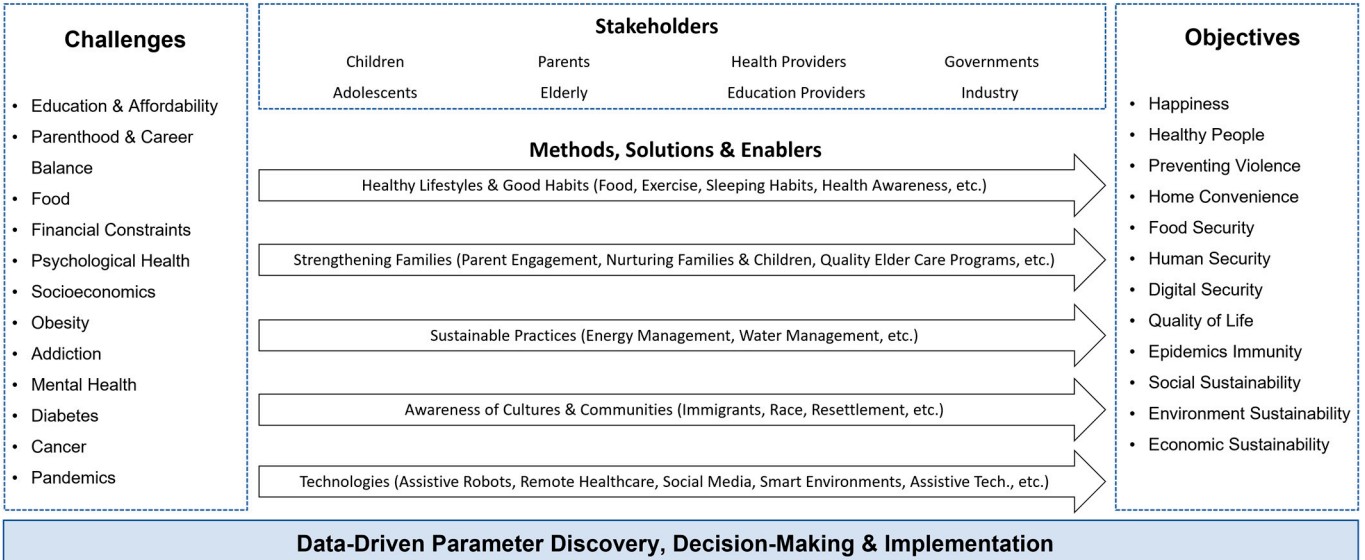

**Figure 25.** Smart Families & Homes Framework.

## 7. Conclusions

It is clear that the current efforts on smart home research and products are limited in their scope and do not consider the concept of home, particularly its social dimensions. Homes need to be seen as fundamental units and beacons of society. An individual's character is defined by the values they hold. Society is comprised of individuals and, therefore, an individual's character determines the social character. The roles homes need to play in societies must be investigated and defined such that smart home technologies prioritize nurturing at home human values of sincerity, honesty, integrity, courage, determination, tolerance, compassion, generosity, loyalty, humility, modesty, and democracy, etc. These values should be nurtured in homes, the places that could have, and should have, the highest capacity for positive influence on its inhabitants, as well as the society. Thereby, homes should be developed as a place of love, trust, tolerant relationships, learning, and value building, a beacon of good and repellant of vice.

While it is important that the research on smart homes continues to develop technologies in their current domains of activities, it is vital, even more so, that smart homes research is guided through a holistic understanding of home meanings and functions and is aligned with the discrete and broad smart society objectives. There is a clear gap in understanding, defining, and actualization of the overarching roles of smart homes, the roles of smart homes that would serve the needs of future smart cities and societies.

This paper introduced our data-driven parameter discovery methodology and used it to provide, for the first time, a comprehensive analysis of the families and homes landscape as seen by academics and the general public, using over a hundred thousand research papers and nearly a million tweets. We created a methodology that uses deep learning, natural language processing (NLP), and big data analytics to automatically discover parameters that capture a comprehensive knowledge and design space for smart families and homes that includes social, political, economic, environmental, and other dimensions. The 66 discovered parameters and the knowledge space with hundreds of dimensions were explained by reviewing and referencing over 300 academic articles and tweets.

The knowledge and parameters discovered in this paper can be used to develop a holistic understanding of issues concerning families and homes, thereby facilitating the development of better policies, technologies, solutions, and industries for families and homes, and thus strengthening families and homes and, as a result, empowering sustainable societies worldwide. The parameters detected by Twitter data reveal a very local national

view of families, with parameters such as an emphasis on good companionship, gatherings with grandmothers, and laziness. These could be used to research and develop cultures-specific, intercultural, and intercommunal norms, traditions, and sensitivities in families and homes. Many works relating to families and homes are being carried out under various research umbrellas. Our approach could aid in the consolidation and streamlining of efforts to develop more sustainable technologies for smart homes and societies.

The methodology and analysis developed in this work are extensible and applicable to other subjects and topics, with seemingly limitless applications. For example, the approach makes use of Scopus data, which could be extended to other academic databases in order to develop and discover more comprehensive models of families and homes. Similarly, data sources other than Twitter, such as social media platforms, magazines, and web resources, could be added. One challenge of using social media platforms such as Twitter is that businesses use them to market their products. This has most likely contributed to the overemphasis of home cleaning-related parameters, but it may also reflect demand and thus an acceptable view of families and homes. This will be looked into further in the future in order to address the undesired influence of commercial tweets on the knowledge space.

**Author Contributions:** Conceptualization, E.A., N.J. and R.M.; methodology, E.A., N.J. and R.M.; software, E.A.; validation, E.A., N.J. and R.M.; formal analysis, E.A., N.J., R.M. and S.S.; investigation, E.A., N.J., R.M. and S.S.; resources, R.M. and S.S.; data curation, E.A.; writing—original draft preparation, E.A., N.J. and R.M.; writing—review and editing, R.M. and S.S.; visualization, E.A. and N.J.; supervision, R.M. and S.S.; project administration, R.M.; funding acquisition, R.M. All authors have read and agreed to the published version of the manuscript.

**Funding:** The authors acknowledge with thanks the technical and financial support from the Deanship of Scientific Research (DSR) at the King Abdulaziz University (KAU), Jeddah, Saudi Arabia, under Grant No. RG-11-611-38. The experiments reported in this paper were performed on the Aziz supercomputer at KAU.

**Institutional Review Board Statement:** Not applicable.

**Informed Consent Statement:** Not applicable.

**Data Availability Statement:** The Dataset developed in this work will be provided publicly.

**Acknowledgments:** The work carried out in this paper is supported by the HPC Center at the King Abdulaziz University.

**Conflicts of Interest:** The authors declare no conflict of interest.

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
