# Peer review of "Smart Homes and Families to Enable Sustainable Societies: A Data-Driven Approach for Multi-Perspective Parameter Discovery Using BERT Modelling"

_sustainability, doi:10.3390/su142013534_

Round 1
Reviewer 1 Report
The present paper proposes the relationship between smart homes / families to achieve sustainable societies , applying the use of BERT models for multi-perspective parameter discovery.
We are faced with a highly relevant piece of research, given the importance of the smart home topic. The application of in-depth learning is a coherent strategy and outlines a contribution to the field of application.
It defines a series of objectives that, although coherent with the study area, are addressed and developed excessively for a research paper (which typically ranges from 15 to 20 pages ) In this sense, it lacks synthesis.
From the methodological point of view. It would be a good idea to include a layout identifying the stages addressed in this process. This roadmap could help guide the reader and facilitate knowing which section of this long article they are in.
Reviewer 2 Report
The paper presents a new data-driven framework, which is based on the Bert model. The proposed framework outputs some key words, which can be used for grouping the existing parameters that capture a comprehensive knowledge and design space of academic research articles and tweets.
The strong points of the paper are described below:
i ) The proposed framework is novel and seems to have good performance results.
ii) It is the first work that seems to combine two different daily life "variables" families and homes landscape.
iii) Also, this work is innovative as it is the first that combines the information from academic research papers with daily popular tools, like twitter.
On the other hand, there are some important points of the paper that need to be optimized in order to be published.
i) Your framework is based on the BERTopic framework. On the other hand, your work should include a small description of this framework (Section 3) in order to offer a full description of the used frameworks. After the description the
ii) The temporal analysis of the Section 4.8 should be merged with the corresponding sections that describe the proposed frameworks. This way will group all the useful information of the proposed framework with the corresponding results.
iii) The sections that describe technologies should be reformatted in order to be more clear about the grouping. In each one of these sections, you describe the keywords of the group. After that, there is a description of related works (?) / publications, which are not clear to the reader their meaning. Here the writers should reformat the sections with the parameters and they need to change it offering more concrete results.
iv) Last, there are some typos, which the writers need to correct:
- " Section 3.6 and 3.6 discuss" ==> ????
- Line 170. The sentence is wrong.
- Line 375. You refer to Section 2, which was the previous one.
Based on the above, we propose a major revision to the paper in order to be more clear to the reader its interesting content.
Reviewer 3 Report
This paper introduces a data-driven parameter discovery methodology to provide analysis of the families and homes using over a hundred thousand research papers and nearly a million tweets. A methodology using deep learning, natural language processing (NLP), and big data analytics methods were applied to automatically discover grouped parameters. This is an interesting paper as it has the potential to provide the knowledge and parameters to develop a holistic understanding of matters related to families and homes. Here are some suggestions that might be helpful for the authors to improve their work:
1. What does figure 1 serve in this paper? Explanation for figure 1 might be missing.
2. Fig. 3 and 4 are not histograms. A histogram should have range value as x-axis and frequency as y-axis.
3. The c-TF-IDF, which is defined in equation (1), is essential to understand the following analysis. Therefore, more explanation might be needed for the explanation.
4. It’s not clear to me how each step, from key words to parameters, and from parameters to Macro are generated. A flowchart showing the entire process might be helpful. For example, how do you decide the keywords under one parameter? And how do you decide what parameters should be in on Macro. You might mention it somewhere in the paper but it would be helpful if you have the flowchart to guide the readers.

Reviewer 4 Report
This article offers a methodology that is said to be able to automatically discover parameters that capture appropriate knowledge and design space of smart families and homes through a review of the literature.
My comments are as follows:
1) Abstract: The abstract as a whole is somewhat vague, with a few claims that may or may not be accurate or acceptable. The methodology in the abstract is unclear - BERT modelling is mentioned in the title but not in the abstract. The results and discussion provided in the abstract are vague, as evidenced by the keywords, many of which are not mentioned in the abstract. The abstract introduction can be shortened [Lines 12-23].
2) Introduction: I'd recommend that the authors rearrange the Introduction section beginning with Line 114; the significance and novelty of the study should be brief yet concise and clear, and can be provided at the end of the Introduction section, while other sections (Sections 1.4 and 2.1) can be removed. The research gap should be provided before stating the objectives/importance of the current study. An important detail is missing at the beginning of Line 170, Figure 1?
3) Methodology: Please indicate why Scopus is being used rather than the Web of Science (WOS) to draw academic perspectives. This is because WOS journals are more stringent in the peer-review process in order to publish high-quality research papers.
4) I’d suggest the authors tabulate the relevant details of Section 4 [particularly between Lines 678 – 1632] rather than providing them in paragraphs. This will help improve the readability of the script.
5) Discussion may be valid if the authors can provide evidence to support their findings from the previous. There is not one literature being cited in the entire Discussion section [Lines 2016 – 2104]. As a result, the current Discussion section is unsuitable for an academic paper.
6) The overall presentation of the script could be improved; some figures are unclear (e.g., Figures 1, 5, and 7) and there are numerous typos throughout the manuscript.
Round 2
Reviewer 4 Report
Most of the comments have been addressed by the authors. The revised script, however, is still too lengthy for an article, and the content in the Introduction and Results sections should be drastically reduced.
Author Response
Dear Reviewer,
Thank you very much for your comments. We have now revised the manuscript according to your comments and have shortened the Introduction and Results section.
We hope that the revised manuscript will meet the reviewer’s satisfaction.